# REASSESSING LAYER PRUNING IN LLMS: NEW INSIGHTS AND METHODS

## ABSTRACT

Although large language models (LLMs) have achieved remarkable success across various domains, their considerable scale necessitates substantial computational resources, posing significant challenges for deployment in resource-constrained environments. Layer pruning, as a simple yet effective compression method, removes layers of a model directly, reducing computational overhead. However, what are the best practices for layer pruning in LLMs? Are sophisticated layer selection metrics truly effective? Does the LoRA (Low-Rank Approximation) family, widely regarded as a leading method for pruned model fine-tuning, truly meet expectations when applied to post-pruning fine-tuning? To answer these questions, we dedicate thousands of GPU hours to benchmarking layer pruning in LLMs and gaining insights across multiple dimensions. Our results demonstrate that a simple approach, i.e., pruning the final 25% of layers followed by fine-tuning the `lm_head` and the remaining last three layer, yields remarkably strong performance. Following this guide, we prune Llama-3.1-8B-It and obtain a model that outperforms many popular LLMs of similar size, such as ChatGLM2-6B, Vicuna-7B-v1.5, Qwen1.5-7B and Baichuan2-7B. We release the optimal model weights on Huggingface[1], and the code is available on GitHub[2].

## 1 INTRODUCTION

In recent years, large language models (LLMs) have achieved unprecedented success in many fields, such as text generation (Achiam et al., 2023; Touvron et al., 2023), semantic analysis (Deng et al., 2023; Zhang et al., 2023b) and machine translation (Zhang et al., 2023a; Wang et al., 2023). However, these achievements come with massive resource consumption, posing significant challenges for deployment on resource-constrained devices. To address these challenges, numerous techniques have been developed to create more efficient LLMs, including pruning (Ma et al., 2023a; Sun et al., 2023), knowledge distillation (Xu et al., 2024; Gu et al., 2024), quantization (Lin et al., 2024; Liu et al., 2023), low-rank factorization (Saha et al., 2023; Zhao et al., 2024a), and system-level inference acceleration (Shah et al., 2024; Lee et al., 2024).

Among these methods, pruning has emerged as a promising solution to mitigate the resource demands of LLMs. By selectively removing redundant patterns—such as parameters (Sun et al., 2023), attention heads (Ma et al., 2023a) and layers (Men et al., 2024)—pruning aims to slim down the model while maintaining its original performance as much as possible. Among different types of pruning, layer pruning (Kim et al., 2024; Siddiqui et al., 2024) has garnered particular interest due to its direct impact on pruning the model's depth, thereby decreasing both computational complexity and memory usage. Additionally, thanks to the nice structure of the existing LLMs such as Llama (Dubey et al., 2024), whose transformer blocks have the exactly same dimension of input and output, layer pruning becomes a straightforward and simple solution. Therefore, in this paper, we focus on layer pruning. Unlike existing studies (Men et al., 2024; Yang et al., 2024b; Chen et al., 2024; Zhong et al., 2024; Liu et al., 2024b) that aim to propose various sophisticated pruning methods, we take a step back and focus on the following questions:

---

[1]https://huggingface.co/anonymousICLR/Llama-3.1-6.3B-It-Alpaca and https://huggingface.co/anonymousICLR/Llama-3.1-6.3B-It-Dolly/

[2]https://anonymous.4open.science/r/Navigation-LLM-layer-pruning-DEB7

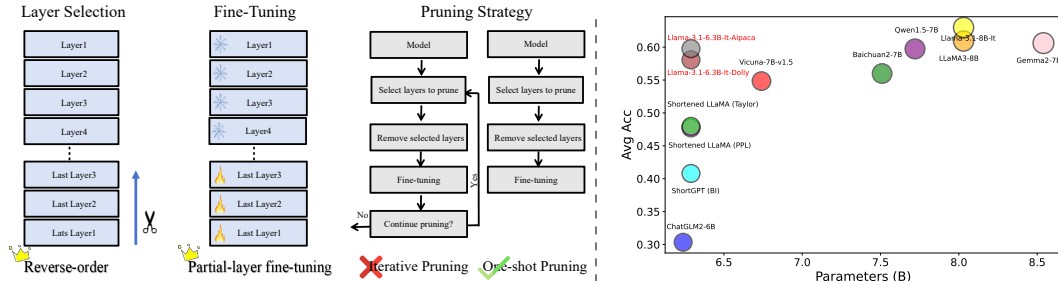

Figure 1: Insights for best practices (left) and the pruned models (right). Insights: 1) Prune from the tail. 2) Fine-tune the last few layers (instead of using LoRA). 3) Iterative pruning benefits rarely. Pruned models: Llama-3.1-6.3B-It-Alpaca and Llama-3.1-6.3B-It-Dolly achieve a good trade-off between performance and model size, as they are positioned in the top left corner.

**Q1.** *Layer Selection:* Are fancy metrics essential for identifying redundant layers to prune?

**Q2.** *Fine-Tuning:* Is the LoRA family the best choice for post-pruning fine-tuning?

**Q3.** *Pruning Strategy:* Will iterative pruning outperform one-shot pruning?

To answer the aforementioned questions, we spent thousands of GPU hours to benchmark layer pruning, conducting extensive experiments across **7** layer selection metrics, **4** state-of-the-art open-source LLMs, **6** fine-tuning methods, **5** pruning strategies on **10** common datasets. From these efforts, we have developed a practical list of key insights for LLM layer pruning in Figure 1:

1). **Reverse-order pruning is simple yet effective**, i.e., simply pruning the last several layers performs better than many complex pruning metrics (Kim et al., 2024; Men et al., 2024) .

2). **LoRA performs worse than expected**, i.e., LoRA, the most commonly used fine-tuning methods in existing pruning approaches (Sun et al., 2023; Ma et al., 2023b; Kim et al., 2024; Men et al., 2024), is **not** the best choice for post-pruning performance recovery. In contrast, freezing the other layers and fine-tuning only the last few remaining layers and *lm_head*, also known as *partial-layer fine-tuning*, can achieve higher accuracy while reducing the training time. The result is unique to layer pruning since LoRA and partial-layer fine-tuning perform similarly as Table 3 in full-model fine-tuning.

3). **Iterative pruning offers no benefit**, i.e., considering both training costs and performance gains, iterative pruning, where layers are removed step-by-step, fails to beat the one-shot pruning, where a single cut is made.

In addition to the above practices, we also conduct sensitivity analyses on the number of calibration samples, the choice of Supervised Fine-Tuning (SFT) datasets and various pruning rates for LLM layer pruning. We find that the number of calibration samples affects the performance of data-driven pruning methods, highlighting the importance of considering performance stability as a key criterion when evaluating the quality of pruning metrics. Similarly, we discover that fine-tuning with different SFT datasets significantly impacts the performance of pruned models. This suggests the need for further exploration of the most suitable datasets for fine-tuning. Finally, we apply our insights and practices to prune Llama-3.1-8B-Instruct (Dubey et al., 2024), obtaining **Llama-3.1-6.3B-It-Alpaca** and **Llama-3.1-6.3B-It-Dolly**, as shown in Figure 1. These pruned models require significantly fewer training tokens but outperform several popular community LLMs of similar size, such as ChatGLM2-6B (GLM et al., 2024), Vicuna-7B-v1.5 (Zheng et al., 2024), Qwen1.5-7B (Yang et al., 2024a) and Baichuan2-7B (Baichuan, 2023). We hope our work will help guide future efforts in LLM layer pruning and inform best practices for deploying LLMs in real-world applications. In a nutshell, we make the following contributions:

- *Comprehensive Benchmarking:* We conduct an extensive evaluation of layer selection metrics, fine-tuning methods, and pruning strategies, providing practical insights into effective pruning techniques based on thousands of GPU hours across multiple datasets.

- *Novel Best Practices:* We identify reverse-order as a simple and effective layer selection metric, find that partial-layer fine-tuning outperforms LoRA-based techniques, and demonstrate that one-shot pruning is as effective as iterative pruning while reducing training costs.

- *Optimized Pruned LLMs:* We release **Llama-3.1-6.3B-It-Alpaca** and **Llama-3.1-6.3B-It-Dolly**, which are obtained through direct pruning of the Llama-3.1-8B-Instruct. Our pruned models require up to $10^6\times$ fewer training tokens compared to training from scratch, while still comparing favorably to various popular community LLMs of similar size, such as ChatGLM2-6B (GLM et al., 2024), Vicuna-7B-v1.5 (Zheng et al., 2024), Qwen1.5-7B (Yang et al., 2024a) and Baichuan2-7B (Baichuan, 2023).

## 2 RELATED WORK

**LLM Layer Pruning.** LLM layer pruning is a technique used to reduce the number of layers in LLMs, aiming to lower computational costs without significantly degrading performance. Specifically, it evaluates the contribution of each layer to the model's overall performance, using criteria such as gradients, activation values, parameter weights, or the layer's influence on the loss function. Layers that contribute the least are then pruned to reduce complexity. For example, LaCo (Yang et al., 2024b) achieves rapid model size reduction by folding subsequent layers into the previous layer, effectively preserving the model structure. Similarly, MKA (Liu et al., 2024b) uses manifold learning and the Normalized Pairwise Information Bottleneck measure (Tishby et al., 2000) to identify the most similar layers for merging. ShortGPT (Men et al., 2024) uses Block Influence (BI) to measure the importance of each layer in LLMs and remove layers with low BI scores. Kim et al. (2024) utilize Magnitude, Taylor and Perplexity (PPL) to evaluate the significance of each layer.

**Differences from Traditional Layer Pruning.** Unlike traditional Deep Neural Networks (Szegedy et al., 2014; Simonyan & Zisserman, 2015; He et al., 2015; Dosovitskiy et al., 2021; Liu et al., 2021) (DNNs), typically trained for a single, specific task, LLMs are designed to handle a wide range of tasks and are structured with billions of parameters. These differences in model scale and task complexity fundamentally alter the challenges associated with layer pruning. For example, in traditional DNN layer pruning (Chen & Zhao, 2018; Wang et al., 2019; Lu et al., 2022; Tang et al., 2023; Guenter & Sideris, 2024), assessing the importance of each layer is relatively straightforward, as it is tied to a single task. In contrast, the parameters of LLMs are optimized across diverse tasks, complicating the evaluation of layer importance. Furthermore, traditional DNN pruning commonly involves full parameter fine-tuning after pruning, while LLMs often employ Parameter-Efficient Fine-Tuning (PEFT) techniques (Hu et al., 2021; Meng et al., 2024; Zhao et al., 2024b; Dettmers et al., 2024) such as Low-Rank Approximation (LoRA) (Hu et al., 2021) to accommodate their massive parameter space. Consequently, traditional DNN pruning methods may not adequately address the unique challenges posed by LLMs, highlighting the need for specialized pruning strategies.

**Exploration of LLM Pruning.** Although recent research focuses on developing sophisticated pruning methods (Kim et al., 2024; Ma et al., 2023a; Men et al., 2024; Liu et al., 2024c;b; Yang et al., 2024b; Zhong et al., 2024), few studies (Jaiswal et al., 2023; Williams & Aletras, 2024; Muralidharan et al., 2024) take a step back and revisit existing LLM pruning techniques. For example, Jaiswal et al. (2023) re-evaluate the effectiveness of existing state-of-the-art pruning methods with PPL. Williams & Aletras (2024) systematically investigate how the calibration dataset impacts the effectiveness of model compression methods. Muralidharan et al. (2024) develop a set of practical practices for LLMs that combine layer, width, attention and MLP pruning with knowledge distillation-based retraining. However, these methods either do not consider layer pruning or lack a comprehensive comparison. In contrast, we systematically validate different layer selection metrics, fine-tuning techniques, and pruning strategies to provide a thorough evaluation.

## 3 BACKGROUND AND NOTATION

### 3.1 PROBLEM FORMULATION FOR LAYER PRUNING

An LLM $\mathcal{M}$ consists of multiple Transformer layers $L = \{l_1, l_2, \cdots, l_n\}$, each containing a pair of multi-head attention and feed-forward network modules:

$$\mathcal{M} = l_1 \circ l_2 \cdots \circ l_n, \tag{1}$$

Layer pruning aims to find a subset of layers $L' \subseteq L$ such that the pruned model $\mathcal{M}'$ maintains acceptable performance while reducing the model's complexity, which can be formalized as:

$$
\begin{aligned}
\text{Minimize} \quad & \mathcal{C}\left(\mathcal{M}'\right), \\
\text{s.t.} \quad & P\left(\mathcal{M}'\right) \geq \alpha \times P(\mathcal{M}), L' \subseteq L,
\end{aligned} \tag{2}
$$

where $\mathcal{C}\left(\mathcal{M}'\right)$ denotes the complexity of the pruned model, which can be quantified in terms of the number of parameters, FLOPs, or inference time, etc. $\alpha$ is a hyperparameter (e.g., $\alpha = 0.9$) that defines the acceptable performance degradation. $P(\cdot)$ represents the performance on given tasks. Numerous methods have proposed various metrics to identify and prune unimportant layers. Herein, we include 7 popular metrics:

**Random Selection.** For the random selection baseline, we randomly select several layers to prune.

**Reverse-order**. This metric (Men et al., 2024) posits that importance is inversely proportional to the sequence order. It assigns lower importance scores to the deeper layers and prune them.

**Magnitude.** It was first introduced by Li et al. (2016) and subsequently adopted by Kim et al. (2024), which assumes that weights exhibiting smaller magnitudes are deemed less informative. Following Kim et al. (2024), we compute $I_{\text{Magnitude}}^n = \sum_k ||W_k^n||_p$, where $W_k^n$ denotes the weight matrix of operation $k$ within the $n$-th transformer layer. In this paper, we uniformly set $p = \{1, 2\}$. As a result, we term these methods as **Magnitude-l1** and **Magnitude-l2**.

**Taylor.** For a given calibration dataset $D$, the significance of removing weight parameters is indicated by the change in training loss $\mathcal{L} := |\mathcal{L}(W_k^n, D) - \mathcal{L}(W_k^n = 0, D)| \approx |\frac{\partial \mathcal{L}(D)}{\partial W_k^n} W_k^n|$. Following Ma et al. (2023a); Kim et al. (2024), we omit the second-order derivatives in this assessment. Then we define the Taylor score of the $n$-th transformer layer as $I_{\text{Taylor}}^n = \sum_k |\frac{\partial \mathcal{L}(D)}{\partial W_k^n} W_k^n|$.

**PPL.** Following Kim et al. (2024), we remove a single layer and assess its impact on the perplexity of the pruned model using the calibration dataset $D$. We then prune those layers that lead to a smaller degradation of the PPL.

**BI.** Men et al. (2024) introduce a metric called Block Influence as an effective indicator of layer importance. Specifically, the BI score of the $i$-th layer can be calculated as follows:

$$
\text{BI}_i = 1 - \mathbb{E}_{X,t} \frac{X_{i,t}^T X_{i+1,t}}{\|X_{i,t}\|_2 \|X_{i+1,t}\|_2}, \tag{3}
$$

where $X_i$ denotes the input of the $i$-th layer and $X_{i,t}$ is the $t$-th row of $X_i$.

### 3.2 EVALUATION AND DATASETS

To assess the performance of the model, we follow the evaluation of Ma et al. (2023a) to perform zero-shot task classification on 8 common sense reasoning datasets using the lm-evaluation-harness (Gao et al., 2023) package: MMLU (Hendrycks et al., 2021), CMMLU (Li et al., 2023), PIQA (Bisk et al., 2020), HellaSwag (Zellers et al., 2019), WinoGrande (Sakaguchi et al., 2021), ARC-easy (Clark et al., 2018), ARC-challenge (Clark et al., 2018) and OpenbookQA (Mihaylov et al., 2018). Additionally, we evaluate the model using perplexity on the WikiText2 (Merity et al., 2016) and Penn Treebank (PTB) (Marcus et al., 1993) datasets. For the PPL metric, we follow (Ma et al., 2023a; Muralidharan et al., 2024) and use WikiText2 for calculation. Following (Ma et al., 2023a), we randomly select 10 samples from BookCorpus (Zhu et al., 2015) to compute Taylor and BI, truncating each sample to a sequence length of 128. Unless otherwise specified, we utilize the Alpaca-cleaned (Taori et al., 2023) with LoRA to recover the performance. Uniformly, we set the training epoch to 2 and batch size to 64. All experiments are conducted on 2 NVIDIA A100 GPUs with 40 GB of memory and 4 NVIDIA RTX A5000 GPUs with 24 GB of memory.

## 4 AN EMPIRICAL EXPLORATION OF LLM LAYER PRUNING

This paper aims to contribute to the community the best practice of layer pruning such that practitioners can prune an LLM to an affordable size and desired performance with minimal exploration effort. Specifically, we will expand from three aspects: First, we explore which metric is most

Table 1: Zero-shot performance of the pruned models (25% pruning rate, fine-tuning using LoRA). "Avg Acc" denotes the average accuracy calculated among eight datasets. The best results are marked in **boldface**, and the sub-optimal ones are underlined.

| Model | Metric | Benchmarks | | | | | | | | Avg Acc |
|---|---|---|---|---|---|---|---|---|---|---|
| | | PIQA | HellaSwag | OpenbookQA | ARC-e | ARC-c | MMLU | CMMLU | WinoGrande | |
| Vicuna-7B-v1.5 | Dense | 0.7720±0.0098 | 0.5642±0.0049 | 0.3300±0.0210 | 0.7555±0.0088 | 0.4326±0.0145 | 0.4858±0.0040 | 0.3518±0.0044 | 0.6953±0.0129 | 0.5484 |
| | Reverse-order | 0.7171±0.0105 | **0.5005±0.0050** | 0.2608±0.0198 | 0.6221±0.0099 | **0.3848±0.0142** | **0.4737±0.0041** | **0.3417±0.0044** | **0.6267±0.0136** | **0.4909** |
| | Random | 0.5223±0.0117 | 0.2607±0.0044 | 0.1380±0.0154 | 0.2614±0.0090 | 0.2176±0.0121 | 0.2295±0.0035 | 0.2500±0.0040 | 0.4672±0.0140 | 0.2933 |
| | PPL | **0.7361±0.0103** | 0.4734±0.0050 | **0.2760±0.0200** | **0.6705±0.0096** | 0.3456±0.0139 | 0.2943±0.0038 | 0.2569±0.0041 | 0.5896±0.0138 | 0.4553 |
| | Magnitude-l1 | 0.5299±0.0116 | 0.2586±0.0044 | 0.1440±0.0157 | 0.2609±0.0090 | 0.2253±0.0122 | 0.2297±0.0035 | 0.2514±0.0040 | 0.4893±0.0140 | 0.2986 |
| | Magnitude-l2 | 0.5256±0.0117 | 0.2578±0.0044 | 0.1340±0.0152 | 0.2622±0.0090 | 0.2108±0.0119 | 0.2295±0.0035 | 0.2515±0.0040 | 0.4838±0.0140 | 0.2944 |
| | BI | 0.6910±0.0108 | 0.3987±0.0049 | 0.2100±0.0182 | 0.5829±0.0101 | 0.2654±0.0129 | 0.2389±0.0036 | 0.2513±0.0040 | 0.5036±0.0141 | 0.3927 |
| | Taylor | 0.5250±0.0117 | 0.2581±0.0044 | 0.1360±0.0153 | 0.2584±0.0090 | 0.2048±0.0118 | 0.2318±0.0036 | 0.2526±0.0040 | 0.4972±0.0141 | 0.2955 |
| Qwen1.5-7B | Dense | 0.7845±0.0096 | 0.5785±0.0049 | 0.3160±0.0208 | 0.7125±0.0093 | 0.4053±0.0143 | 0.5967±0.0039 | 0.7277±0.0039 | 0.6575±0.0133 | 0.5973 |
| | Reverse-order | 0.6942±0.0107 | **0.4444±0.0050** | 0.2280±0.0188 | 0.5143±0.0103 | **0.3302±0.0137** | **0.5101±0.0041** | **0.7171±0.0040** | 0.5912±0.0138 | **0.5037** |
| | Random | 0.5408±0.0116 | 0.2682±0.0044 | 0.1240±0.0148 | 0.2630±0.0090 | 0.2039±0.0118 | 0.2366±0.0076 | 0.2457±0.0040 | 0.4807±0.0140 | 0.2954 |
| | PPL | 0.7089±0.0106 | 0.4195±0.0049 | 0.2240±0.0187 | 0.5960±0.0101 | 0.2944±0.0133 | 0.2457±0.0036 | 0.2552±0.0041 | 0.5185±0.0140 | 0.4078 |
| | Magnitude-l1 | 0.6578±0.0111 | 0.3989±0.0049 | 0.2040±0.0180 | 0.5244±0.0102 | 0.2901±0.0133 | 0.2574±0.0037 | 0.2541±0.0041 | 0.5249±0.0140 | 0.3890 |
| | Magnitude-l2 | 0.5903±0.0115 | 0.3657±0.0048 | 0.1640±0.0166 | 0.4630±0.0102 | 0.2381±0.0124 | 0.2502±0.0037 | 0.2513±0.0040 | 0.5312±0.0140 | 0.3567 |
| | BI | **0.7220±0.0105** | 0.4190±0.0049 | **0.2440±0.0192** | **0.5972±0.0101** | 0.2671±0.0129 | 0.2456±0.0036 | 0.2536±0.0040 | 0.5383±0.0140 | 0.4190 |
| | Taylor | 0.6970±0.0107 | 0.4284±0.0049 | 0.2060±0.0181 | 0.5160±0.0103 | 0.3140±0.0136 | **0.5231±0.0041** | 0.6079±0.0043 | **0.6046±0.0137** | 0.4871 |
| Gemma2-2B-It | Dense | 0.7867±0.0096 | 0.5367±0.0050 | 0.3560±0.0214 | 0.8085±0.0081 | 0.5111±0.0146 | 0.5687±0.0039 | 0.4499±0.0045 | 0.6961±0.0129 | 0.5892 |
| | Reverse-order | 0.7029±0.0107 | 0.4529±0.0050 | 0.2660±0.0198 | 0.6343±0.0099 | **0.3763±0.0142** | **0.5261±0.0040** | **0.4117±0.0045** | 0.6551±0.0134 | 0.5032 |
| | Random | 0.7307±0.0104 | 0.4462±0.0050 | 0.2860±0.0202 | 0.6852±0.0095 | 0.3422±0.0139 | 0.3452±0.0040 | 0.2893±0.0042 | 0.5833±0.0139 | 0.4635 |
| | PPL | 0.7454±0.0102 | **0.4611±0.0050** | 0.2940±0.0204 | 0.7008±0.0094 | 0.3609±0.0140 | 0.3503±0.0040 | 0.2838±0.0042 | 0.5825±0.0139 | 0.4724 |
| | Magnitude-l1 | **0.7481±0.0101** | 0.4530±0.0050 | **0.3040±0.0206** | **0.7239±0.0092** | 0.3729±0.0141 | 0.2703±0.0037 | 0.2514±0.0040 | 0.5596±0.0140 | 0.4604 |
| | Magnitude-l2 | 0.7225±0.0104 | 0.4245±0.0049 | 0.2380±0.0191 | 0.6561±0.0097 | 0.3038±0.0134 | 0.2413±0.0036 | 0.2258±0.0041 | 0.5493±0.0140 | 0.4202 |
| | BI | 0.6921±0.0108 | 0.4272±0.0049 | 0.2700±0.0199 | 0.6511±0.0098 | 0.3703±0.0141 | 0.4968±0.0040 | 0.3851±0.0045 | **0.6661±0.0133** | 0.4948 |
| | Taylor | 0.7002±0.0107 | 0.4541±0.0050 | 0.3020±0.0206 | 0.6359±0.0099 | 0.3695±0.0141 | **0.5431±0.0040** | 0.4048±0.0045 | 0.6488±0.0134 | **0.5073** |
| Llama-3.1-8B-It | Dense | 0.8003±0.0093 | 0.5910±0.0049 | 0.3380±0.0212 | 0.8182±0.0079 | 0.5179±0.0146 | 0.6790±0.0039 | 0.5552±0.0045 | 0.7395±0.0123 | 0.6299 |
| | Reverse-order | 0.7002±0.0107 | 0.4010±0.0049 | **0.2940±0.0204** | 0.6170±0.0100 | 0.3985±0.0143 | **0.6342±0.0039** | **0.5449±0.0045** | 0.6243±0.0136 | **0.5268** |
| | Random | 0.5653±0.0116 | 0.2886±0.0045 | 0.1400±0.0155 | 0.3169±0.0095 | 0.1860±0.0114 | 0.2275±0.0035 | 0.2559±0.0041 | 0.5075±0.0141 | 0.3110 |
| | PPL | **0.7628±0.0099** | 0.4931±0.0050 | 0.2640±0.0197 | **0.7290±0.0091** | 0.3805±0.0142 | 0.3367±0.0040 | 0.2724±0.0041 | 0.5793±0.0139 | 0.4772 |
| | Magnitude-l1 | 0.5408±0.0116 | 0.2634±0.0044 | 0.1360±0.0153 | 0.2845±0.0093 | 0.2014±0.0117 | 0.2504±0.0037 | 0.2503±0.0040 | 0.4878±0.0140 | 0.3018 |
| | Magnitude-l2 | 0.5413±0.0116 | 0.2638±0.0044 | 0.1340±0.0152 | 0.2841±0.0093 | 0.2014±0.0117 | 0.2498±0.0036 | 0.2504±0.0040 | 0.4870±0.0140 | 0.3015 |
| | BI | 0.7176±0.0105 | 0.4196±0.0049 | 0.2020±0.0180 | 0.6107±0.0100 | 0.2841±0.0132 | 0.2417±0.0036 | 0.2494±0.0040 | 0.5391±0.0140 | 0.4080 |
| | Taylor | 0.7138±0.0105 | **0.4964±0.0050** | 0.2740±0.0200 | 0.6848±0.0095 | **0.4181±0.0144** | 0.2861±0.0038 | 0.2504±0.0040 | **0.7135±0.0127** | 0.4796 |

effective for identifying unimportant layers, helping researchers make informed choices. Then, we investigate which fine-tuning method most effectively restores model performance after pruning. Finally, we delve deeper into various pruning strategies and want to answer whether iterative pruning will outperform one-shot pruning.

## 4.1 ARE FANCY METRICS ESSENTIAL FOR IDENTIFYING REDUNDANT LAYERS TO PRUNE?

The first question is to find the most "redundant" layers to prune. As discussed in Section 3.1, there are various metrics for layer selection, which can be as straightforward as reverse-order, or as complicated as BI. However, does a complicated metric always contribute to a better performance? Probably not. We find that a simple metric, i.e., reverse-order, is competitive among these metrics.

Specifically, we conduct comprehensive experiments on Vicuna-7B-v1.5 (Zheng et al., 2024), Qwen1.5-7B (Yang et al., 2024a), Gemma2-2B-Instruct (Team, 2024) and Llama-3.1-8B-Instruct (Dubey et al., 2024). We uniformly prune 8 layers (25% pruning ratio) for Vicuna-7B-v1.5, Qwen1.5-7B and Llama-3.1-8B-Instruct, and 6 layers for Gemma2-2B-Instruct. Experiments with a 50% pruning ratio (12 layers for Gemma2-2B-Instruct and 16 layers for others) are provided in Table A. In the fine-tuning stage, we use LoRA with a rank $d$ of 8 and a batch size of 64, and the AdamW optimizer. The learning rate is set to $1 \times 10^{-5}$ with 100 warming steps.

**Results.** As shown in Table 1, we find that the reverse-order metric delivers stable and superior results across various models under the 25% pruning rate, making it a reliable choice for pruning. On average, it outperforms the second-best PPL metric by $5.30\%$ across four models. The result also holds for the 50% pruning rate, as shown in Table A. We hope our insights can help researchers make informed choices when selecting the most suitable pruning metrics for their specific models.

> **Insight #1: The reverse-order are simple yet foolproof metrics for pruning, providing stable and reliable results across different models and pruning rates.**

Table 2: Zero-shot performance of pruned models using various fine-tuning methods under 25% pruning rate (using reverse-order). "Avg Acc" denotes the average accuracy calculated among eight datasets. The best results are marked in **boldface**, and the sub-optimal ones are underlined.

| Model | Method | Layer | Benchmarks | | | | | | | | Avg Acc |
|---|---|---|---|---|---|---|---|---|---|---|---|
| | | | PIQA | HellaSwag | OpenbookQA | ARC-e | ARC-c | MMLU | CMMLU | WinoGrande | |
| Vicuna-7B-v1.5 | LoRA | - | 0.7171±0.0105 | 0.5005±0.0050 | 0.2608±0.0198 | 0.6221±0.0099 | 0.3848±0.0142 | 0.4737±0.0041 | 0.3417±0.0044 | 0.6267±0.0136 | 0.4909 |
| | QLoRA | - | 0.6649±0.0110 | 0.4057±0.0049 | 0.2700±0.0199 | 0.5345±0.0101 | 0.3439±0.0139 | 0.4809±0.0041 | 0.3473±0.0044 | 0.6014±0.0138 | 0.4561 |
| | Partial-layer | *lm_head only* | 0.7057±0.0106 | 0.4865±0.0050 | 0.2880±0.0203 | 0.6301±0.0099 | 0.4010±0.0143 | 0.4819±0.0041 | 0.3520±0.0044 | 0.6156±0.0137 | 0.4951 |
| | | *lm_head+last layer* | 0.7155±0.0105 | 0.5054±0.0050 | 0.2900±0.0203 | 0.6511±0.0098 | **0.4113±0.0144** | 0.4831±0.0041 | 0.3538±0.0044 | **0.6283±0.0136** | 0.5048 |
| | | *lm_head+last two layers* | 0.7214±0.0105 | 0.5060±0.0050 | **0.3020±0.0206** | **0.6532±0.0098** | 0.4002±0.0143 | 0.4858±0.0041 | 0.3530±0.0044 | 0.6267±0.0136 | **0.5060** |
| | | *lm_head+last three layers* | **0.7247±0.0104** | **0.5103±0.0050** | 0.2960±0.0204 | 0.6528±0.0098 | 0.3985±0.0143 | **0.4870±0.0040** | 0.3544±0.0044 | 0.6219±0.0136 | 0.5057 |
| Qwen1.5-7B | LoRA | - | 0.6942±0.0107 | 0.4444±0.0050 | 0.2280±0.0188 | 0.5143±0.0103 | 0.3302±0.0137 | 0.5101±0.0041 | 0.7171±0.0040 | 0.5912±0.0138 | 0.5037 |
| | QLoRA | - | 0.6697±0.0107 | 0.4028±0.0049 | 0.2400±0.0191 | 0.4760±0.0102 | 0.2969±0.0134 | 0.4797±0.0041 | 0.6914±0.0041 | 0.5825±0.0139 | 0.4799 |
| | Partial-layer | *lm_head only* | 0.7149±0.0105 | 0.4735±0.0050 | 0.2460±0.0193 | 0.5497±0.0102 | 0.3524±0.0140 | 0.5467±0.0040 | 0.7276±0.0039 | 0.5967±0.0138 | 0.5259 |
| | | *lm_head+last layer* | 0.7220±0.0105 | 0.4850±0.0050 | 0.2440±0.0192 | 0.5690±0.0102 | 0.3549±0.0140 | 0.5719±0.0040 | **0.7283±0.0039** | 0.6275±0.0136 | 0.5378 |
| | | *lm_head+last two layers* | 0.7214±0.0105 | 0.4915±0.0050 | 0.2540±0.0195 | 0.5783±0.0101 | 0.3584±0.0140 | 0.5734±0.0040 | 0.7275±0.0039 | **0.6298±0.0136** | 0.5418 |
| | | *lm_head+last three layers* | **0.7296±0.0104** | **0.4974±0.0050** | 0.2520±0.0194 | **0.5808±0.0101** | 0.3618±0.0140 | 0.5795±0.0040 | 0.7272±0.0040 | 0.6275±0.0136 | **0.5445** |
| Llama-3.1-8B-It | LoRA | - | 0.7002±0.0107 | 0.4010±0.0049 | 0.2940±0.0204 | 0.6170±0.0100 | 0.3985±0.0143 | 0.6342±0.0039 | 0.5449±0.0045 | 0.6243±0.0136 | 0.5268 |
| | QLoRA | - | 0.6980±0.0107 | 0.3975±0.0049 | 0.3000±0.0205 | 0.6183±0.0100 | 0.3840±0.0142 | 0.6032±0.0039 | 0.5090±0.0045 | 0.6267±0.0136 | 0.5171 |
| | Partial-layer | *lm_head only* | 0.7334±0.0103 | 0.4896±0.0050 | 0.2860±0.0202 | 0.7012±0.0094 | 0.4411±0.0145 | 0.6122±0.0040 | 0.5442±0.0045 | **0.6717±0.0132** | 0.5599 |
| | | *lm_head+last layer* | 0.7350±0.0103 | 0.5107±0.0050 | 0.2940±0.0204 | 0.7193±0.0092 | 0.4531±0.0145 | **0.6630±0.0038** | 0.5526±0.0045 | 0.6582±0.0133 | 0.5732 |
| | | *lm_head+last two layers* | 0.7361±0.0103 | 0.5204±0.0050 | **0.3080±0.0207** | 0.7151±0.0093 | 0.4633±0.0146 | 0.6588±0.0038 | **0.5543±0.0045** | 0.6567±0.0133 | 0.5766 |
| | | *lm_head+last three layers* | **0.7383±0.0103** | **0.5323±0.0050** | **0.3080±0.0207** | **0.7260±0.0092** | **0.4684±0.0146** | 0.6567±0.0038 | 0.5515±0.0045 | **0.6646±0.0133** | **0.5807** |

Table 3: Zero-shot performance of original Llama-3.1-8B-It using LoRA and *lm_head+last three layers*. "Avg Acc" denotes the average accuracy calculated among eight datasets.

| Method | Benchmarks | | | | | | | | Avg Acc |
|---|---|---|---|---|---|---|---|---|---|
| | PIQA | HellaSwag | OpenbookQA | ARC-e | ARC-c | MMLU | CMMLU | WinoGrande | |
| Dense | 0.8003±0.0093 | 0.5910±0.0049 | 0.3380±0.0212 | 0.8182±0.0079 | 0.5179±0.0146 | 0.6790±0.0038 | 0.5552±0.0045 | 0.7395±0.0123 | 0.6299 |
| *lm_head+last three layers* | 0.7998±0.0093 | 0.6057±0.0049 | 0.3520±0.0214 | 0.8186±0.0079 | 0.5316±0.0146 | 0.6784±0.0038 | 0.5522±0.0045 | 0.7316±0.0125 | 0.6337 |
| LoRA | 0.8047±0.0092 | 0.6007±0.0049 | 0.3500±0.0214 | 0.8287±0.0077 | 0.5316±0.0146 | 0.6764±0.0038 | 0.5530±0.0045 | 0.7380±0.0124 | 0.6354 |

## 4.2 Is the LoRA family the best choice for post-pruning fine-tuning?

In previous studies (Kim et al., 2024; Men et al., 2024), LoRA is often used to restore the performance of pruned models. This raises question: Is the LoRA family the best choice for post-pruning fine-tuning? To answer this question, we further use QLoRA (Dettmers et al., 2024) and partial-layer fine-tuning techniques to conduct experiments. We briefly introduce these methods as follows:

**LoRA Fine-tuning.** LoRA is one of the best-performed parameter-efficient fine-tuning paradigm that updates dense model layers using pluggable low-rank matrices (Mao et al., 2024). Specifically, for a pre-trained weight matrix $W_0$, LoRA constrains its update by representing the latter with a low-rank decomposition $W_0 + \Delta W = W_0 + BA$. At the beginning of training, $A$ is initialize with a random Gaussian initialization, while $B$ is initialized to zero. During training, $W_0$ is frozen and does not receive gradient updates, while $A$ and $B$ contain trainable parameters. Then the forward pass can be formalized as:

$$W_0 x + \Delta W x = W_0 x + BA x. \tag{4}$$

**QLoRA Fine-tuning.** QLoRA builds on LoRA by incorporating quantization techniques to further reduce memory usage while maintaining, or even enhancing the performance.

**Partial-layer Fine-tuning.** Compared to LoRA and QLoRA, which inject trainable low-rank factorization matrices into each layer, partial-layer fine-tuning simply freezes the weights of some layers while updating only the specified layers to save computing resources and time (Shen et al., 2021; Ngesthi et al., 2021; Peng & Wang, 2020). Following by the common practice of previous studies (Khan & Fang, 2023), we choose to fine-tune only the later layers that are closer to the output, while keeping the earlier layers, which capture more general features, frozen. Specifically, we use two different fine-tuning strategies: one is to finetune only the model head (*lm_head only*), and the other is to finetune the *lm_head* plus the last layer (*lm_head + last layer*), the last two layers (*lm_head + last two layers*), and the last three layers (*lm_head + last three layers*).

In view of the superiority of the reverse-order metric in Section 4.1, we use it to prune here. For the Vicuna-7B-v1.5, Qwen1.5-7B, and Llama-3.1-8B-Instruct models, we prune 8 layers. For the Gemma2-2B-Instruct model, we prune 6 layers. Subsequently, we utilize LoRA, QLoRA and partial-layer fine-tuning methods to restore performance. We provide more results of fine-tuning with the taylor metric in Table B. In particular, because Gemma2-2B-Instruct employs weight ty-

Table 4: The training cost of fine-tuning the pruned Llama-3.1-8B-Instruct (with 8 layers removed in reverse-order) using different methods on 2 empty NVIDIA RTX A100 GPUs.

| | LoRA | QLoRA | lm_head only | lm_head+last layer | lm_head+last two layers | lm_head+last three layers |
|---|---|---|---|---|---|---|
| Trainable parameters | 15.73M | 15.73M | 525.34M | 743.45M | 961.56M | 1179.68M |
| GPU memory | 45.83G | 14.26G | 39.82G | 42.12G | 44.41G | 48.02G |
| Training time (2 epoch) | 10440.30s | 17249.01s | 6952.92s | 7296.76s | 7616.83s | 7931.36s |

ing (Press & Wolf, 2016) to share the weights between the embedding layer and the softmax layer (*lm_head*), we exclude partial-layer fine-tuning in Gemma2-2B-Instruct. For fine-tuning with LoRA and partial-layer methods, we utilize the AdamW optimizer, while for QLoRA, we opt for the paged_adamw_8bit optimizer. All other hyperparameter settings are the same as in Section 4.1.

**Results.** As shown in the Table 2 and Table B, we find that fine-tuning with QLoRA slightly hurts the performance of pruned models compared to LoRA. Excitingly, the effect of partial-layer fine-tuning is *significantly better* than LoRA, providing a viable new direction for fine-tuning models after pruning. In the ablation study, we compare the performance of LoRA with partial-layer fine-tuning for the full model in Table 3, which shows that partial-layer fine-tuning and LoRA perform similarly. This suggests that the conventional insights for the full model fine-tuning do not hold after pruning, i.e., the structural changes and parameter reduction of the model enable partial layer fine-tuning to adapt more effectively to the new parameter distribution and fully leverage the potential benefits of pruning. When considering fine-tuning methods for LLMs, in addition to performance, the training cost is also a significant factor to take into account. Therefore, we compare the training cost of these fine-tuning methods, including training time, gpu memory and trainable parameters. Specifically, we conduct experiments on 2 empty NVIDIA RTX A100 GPUs using the pruned Llama-3.1-8B-Instruct model (with 8 layers removed in reverse order). Table 4 shows the comparison among these fine-tuning methods. We find that compared to LoRA, partial-layer fine-tuning involves more trainable parameters but maintains comparable GPU usage and achieves faster training time. Additionally, partial-layer fine-tuning outperforms LoRA in effectiveness. In contrast, although QLoRA consumes less GPU memory, it has much longer training time and yields poorer performance. In summary, we conclude that partial-layer fine-tuning is an effective approach to restoring the performance of pruned models when sufficient memory is available.

> **Insight #2: Partial-layer fine-tuning can serve as an alternative to LoRA, achieving better performance recovery for pruned models while reducing training time.**

### 4.3 WILL ITERATIVE PRUNING OUTPERFORM ONE-SHOT PRUNING?

In this subsection, we provide insights into the optimal pruning strategy for LLMs. Although Muralidharan et al. (2024) have explored pruning strategies and concluded that iterative pruning offers no benefit, their study focuses on utilizing knowledge distillation (Hinton, 2015) for performance recovery. In contrast, this paper concentrates on layer pruning with LoRA and partial-layer fine-tuning, thereby broadening the scope of pruning strategies evaluated. We briefly introduce the one-shot pruning and iterative pruning:

**One-shot Pruning.** One-shot pruning scores once and then prune the model to a target prune ratio.

**Iterative Pruning.** Iterative pruning alternately processes the score-prune-update cycle until achieving the target prune ratio.

Specifically, we select Llama-3.1-8B-Instruct and Gemma2-2B-Instruct as the base models. For one-shot pruning, we prune 8 layers from the Llama-3.1-8B-Instruct and 6 layers from the Gemma2-2B-Instruct in a single step, guided by the reverse-order and taylor metrics. For iterative pruning with LoRA, we begin by scoring all layers using these metrics. Subsequently, we set the pruning step to 1 and 4 for Llama-3.1-8B-Instruct, and 1 and 3 for Gemma2-2B-Instruct. After each pruning step, we fine-tune the model with LoRA and merge LoRA weights back into the fine-tuned model. This score-prune-fine-tune-merge cycle is repeated until a total of 8 layers are pruned for Llama-3.1-8B-Instruct and 6 layers for Gemma2-2B-Instruct. For iterative pruning with partial-layer fine-tuning, we fine-tune the model using partial-layer fine-tuning (*lm_head + last three layers*) after

Table 5: Zero-shot performance of pruned models (25% pruning rate) using different pruning strategies. "Avg Acc" denotes the average accuracy calculated among eight datasets. The best results are marked in **boldface**. "1:1:8" refers to an iterative pruning process where 1 layer is pruned at a time, and a total of 8 layers are pruned by the end of the process.

| Fine-tuning Method | Model | Metric | Iteration steps | Benchmarks | | | | | | | | Avg Acc |
|---|---|---|---|---|---|---|---|---|---|---|---|---|
| | | | | PIQA | HellaSwag | OpenbookQA | ARC-e | ARC-c | MMLU | CMMLU | WinoGrande | |
| LoRA | Llama-3.1-8B-It | Reverse-order | one-shot | 0.7002+0.0107 | 0.4010+0.0049 | 0.2940+0.0204 | 0.6170+0.0100 | 0.3985±0.0143 | 0.6342±0.0039 | 0.5449±0.0045 | 0.6243±0.0136 | 0.5268 |
| | | | 1:4:8 | 0.7176±0.0105 | 0.4538±0.0050 | 0.2920±0.0204 | 0.6705±0.0096 | 0.4121±0.0139 | 0.6374±0.0039 | 0.5439±0.0045 | 0.6369±0.0135 | 0.5455 |
| | | | 1:1:8 | 0.7160±0.0105 | 0.4470±0.0050 | 0.2860±0.0202 | 0.6637±0.0097 | 0.4061±0.0144 | 0.6440±0.0039 | 0.5425±0.0045 | 0.6448±0.0135 | **0.5438** |
| | | Taylor | one-shot | 0.7138±0.0105 | 0.4964±0.0050 | 0.2740±0.0200 | 0.6848±0.0095 | 0.4181±0.0144 | 0.2861±0.0038 | 0.2504±0.0040 | 0.7135±0.0127 | 0.4796 |
| | | | 1:4:8 | 0.7149±0.0105 | 0.4991±0.0050 | 0.2480±0.0193 | 0.7071±0.0093 | 0.3951±0.0143 | 0.4676±0.0041 | 0.3480±0.0044 | 0.6709±0.0132 | **0.5063** |
| | | | 1:1:8 | 0.6921±0.0108 | 0.4728±0.0050 | 0.2140±0.0184 | 0.6675±0.0097 | 0.3891±0.0142 | 0.4576±0.0041 | 0.3511±0.0044 | 0.6519±0.0134 | 0.4870 |
| | Gemma2-2B-It | Reverse-order | one-shot | 0.7029±0.0107 | 0.4529±0.0050 | 0.2660±0.0198 | 0.6343±0.0099 | 0.3763±0.0142 | 0.5261±0.0040 | 0.4117±0.0045 | 0.6551±0.0134 | 0.5032 |
| | | | 1:3:6 | 0.6953±0.0107 | 0.4523±0.0050 | 0.2900±0.0203 | 0.6397±0.0099 | 0.3729±0.0141 | 0.5418±0.0040 | 0.4013±0.0045 | 0.6496±0.0134 | **0.5054** |
| | | | 1:1:6 | 0.7067±0.0106 | 0.4476±0.0050 | 0.2660±0.0198 | 0.6305±0.0099 | 0.3746±0.0141 | 0.5143±0.0040 | 0.4066±0.0045 | 0.6559±0.0134 | 0.5003 |
| | | Taylor | one-shot | 0.7002±0.0107 | 0.4541±0.0050 | 0.3020±0.0206 | 0.6359±0.0099 | 0.3695±0.0141 | 0.5431±0.0040 | 0.4048±0.0045 | 0.6488±0.0134 | **0.5073** |
| | | | 1:3:6 | 0.7057±0.0106 | 0.4473±0.0050 | 0.2380±0.0191 | 0.6553±0.0098 | 0.3490±0.0139 | 0.3697±0.0040 | 0.2884±0.0042 | 0.5927±0.0138 | 0.4558 |
| | | | 1:1:6 | 0.7236±0.0104 | 0.4544±0.0050 | 0.2860±0.0202 | 0.6574±0.0097 | 0.3490±0.0139 | 0.4763±0.0041 | 0.3801±0.0045 | 0.6306±0.0136 | 0.4947 |
| Partial-layer | Llama-3.1-8B-It | Reverse-order | one-shot | 0.7383±0.0103 | 0.5323±0.0050 | 0.3080±0.0207 | 0.7260±0.0092 | 0.4684±0.0146 | 0.6567±0.0038 | 0.5515±0.0045 | 0.6646±0.0133 | 0.5807 |
| | | | 1:1:8 | 0.7432±0.0102 | 0.5357±0.0050 | 0.2980±0.0205 | 0.7496±0.0089 | 0.4590±0.0146 | 0.6539±0.0038 | 0.5558±0.0045 | 0.6922±0.0130 | **0.5859** |
| | | Taylor | one-shot | 0.7345±0.0103 | 0.5290±0.0050 | 0.3020±0.0206 | 0.7399±0.0090 | 0.4360±0.0145 | 0.6277±0.0039 | 0.4763±0.0046 | 0.7151±0.0127 | **0.5701** |
| | | | 1:1:8 | 0.6300±0.0113 | 0.3553±0.0048 | 0.1760±0.0170 | 0.5177±0.0103 | 0.2756±0.0131 | 0.2611±0.0037 | 0.2557±0.0041 | 0.5312±0.0140 | 0.3753 |

Table 6: The effect of number of calibration samples on LLM layer pruning. "Avg Acc" denotes the average accuracy calculated among eight datasets. It is worth noting that the layers removed when using 1, 5, and 10 calibration samples are the same, as are the layers removed when using 30 and 50 samples. Therefore, the same data is used in these cases. For more details, please refer to Table D.

| | Verification | PPL on WikiText2 | | PPL on PTB | | Avg Acc | |
|---|---|---|---|---|---|---|---|
| | Metric | BI | Taylor | BI | Taylor | BI | Taylor |
| Calibration Samples | 1 | 51.06 | 65.43 | 90.97 | 94.35 | 0.40 | 0.36 |
| | 5 | 43.54 | 65.43 | 79.34 | 94.35 | 0.43 | 0.36 |
| | 10 | 53.53 | 65.43 | 101.64 | 94.35 | 0.41 | 0.36 |
| | 30 | 50.03 | 55.42 | 88.02 | 77.63 | 0.42 | 0.55 |
| | 50 | 59.73 | 55.42 | 103.19 | 77.63 | 0.41 | 0.55 |

each pruning step, and then repeat the score-prune-fine-tune cycle. To avoid the fine-tuned layers being pruned completely, we set the pruning step size to 1. All hyperparameter settings are the same as in Section 4.1. Experiments with iterative pruning of more layers are provided in Table C.

**Results.** By comparing the results of iterative and one-shot pruning in Table 5 and Table C, we find that unlike traditional CNN pruning, which often yields significant performance improvements through iterative pruning (Tan & Motani, 2020; He & Xiao, 2023), the iterative approach for LLMs may not provide the same benefits and can even lead to performance degradation. We believe that is because too much training causes the model to suffer from catastrophic forgetting (Zhai et al., 2024; Liu et al., 2024a). Figure B visualizes the representational similarity of different pruning strategies. From this, we observe that different pruning strategies yield significantly different representations, highlighting the impact of each strategy on the model's learned features. Besides, iterative pruning requires more computational overhead than one-shot pruning, which is not cost-effective with limited performance gains.

> **Insight #3:** Considering both performance gain and computational overhead, iterative pruning has no benefit.

## 5 SENSITIVITY ANALYSIS

In this section, we conduct sensitivity analyses on the number of calibration samples, the choice of SFT dataset and various pruning rates for LLM layer pruning.

**The effect of number of calibration samples on LLM layer pruning.** It is worth noting that some data-driven layer pruning methods, such as BI and Taylor, rely upon calibration samples to generate layer activations. Therefore, we explore the effect of the number of calibration samples on pruning. Specifically, we calculate BI and Taylor metrics using 1, 5, 10, 30, and 50 calibration samples, prune 8 layers based on these metrics, finetune the pruned Llama-3.1-8B-Instruct models using LoRA, and evaluate their performance through lm-evaluation-harness package. For ease of comparison, we

Table 7: The effect of SFT datasets on LLM layer pruning. "Avg Acc" denotes the average accuracy calculated among eight datasets. The best results are marked in **boldface**.

| Dataset | Benchmarks | | | | | | | | Avg Acc |
|---|---|---|---|---|---|---|---|---|---|
| | PIQA | HellaSwag | OpenbookQA | ARC-e | ARC-c | MMLU | CMMLU | WinoGrande | |
| Dolly-15k | **0.7709±0.0098** | **0.5541±0.0050** | 0.3000±0.0205 | **0.7424±0.0090** | **0.4838±0.0146** | **0.6753±0.0038** | **0.5522±0.0045** | **0.7032±0.0128** | **0.5977** |
| Alpaca-cleaned | 0.7383±0.0103 | 0.5323±0.0050 | **0.3080±0.0207** | 0.7260±0.0092 | 0.4684±0.0146 | 0.6567±0.0038 | 0.5515±0.0045 | 0.6646±0.0133 | 0.5807 |
| MMLU | 0.6012±0.0114 | 0.2714±0.0044 | 0.1700±0.0168 | 0.3430±0.0097 | 0.2457±0.0126 | 0.5888±0.0040 | 0.5266±0.0045 | 0.5856±0.0138 | 0.4165 |

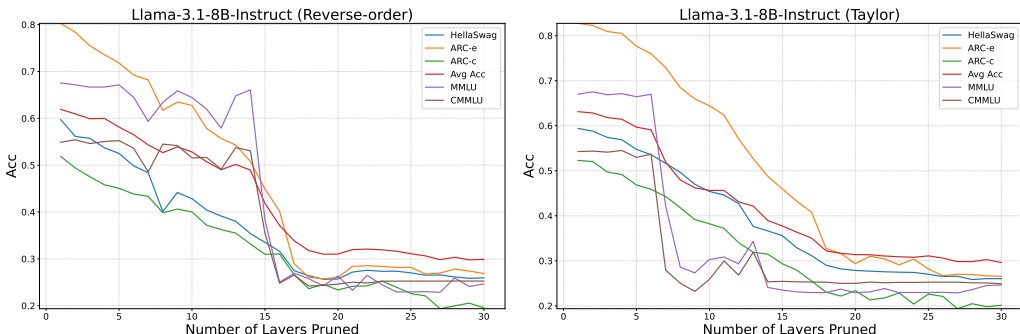

Figure 2: The effect of different pruning rates on LLM layer pruning.

report the average accuracy on 8 datasets in the main text. For more details, see Table D. Besides, we report the model perplexity on the WikiText and Penn Treebank test set. As shown in Table 6, we observe that the pruned models, obtained using varying numbers of calibration samples, do affect the model complexity and zero-shot performance, which suggests that *for data-driven pruning methods, performance stability should also be considered a key criterion when evaluating the quality of pruning technique.*

**The effect of SFT datasets on LLM layer pruning.** In the previous sections, we uniformly utilize Alpaca-cleaned (Taori et al., 2023) to fine-tune the pruned models. Herein, we aim to assess how fine-tuning a pruned model using different SFT datasets affects its performance. Specifically, we conduct experiments using the Reverse-order metric to remove 8 layers from the Llama-3.1-8B-Instruct and fine-tune the pruned model using *lm_head + last three layers* on MMLU (training set) (Hendrycks et al., 2021) and Dolly-15k (Conover et al., 2023). We set the maximum sequence length to 512 for MMLU and 1024 for Dolly-15k. From Table 7, we observe that among these datasets, Dolly-15k achieves the best results, followed by Alpaca-cleaned. This demonstrates that *fine-tuning with different SFT datasets has a significant impact on the performance of pruned models* and suggests further exploration of the most suitable datasets for fine-tuning pruned models.

**The effect of different pruning rates on LLM layer pruning.** We investigate the impact of pruning the LLM at various pruning rates in Figure 2. Specifically, we conduct one-shot pruning on Llama-3.1-8B-Instruct using reverse-order and taylor metrics and evaluate their effects on the model's performance with LoRA. All hyperparameter settings remain consistent with those in Section 4.1. As shown in Figure 2, we observe that as the number of pruned layers increases, the performance of the model on all datasets tends to decrease and eventually converges. However, certain datasets, especially MMLU, CMMLU, and ARC-c, are highly sensitive to layer changes and degrade faster than others. Besides, after cutting off about 16 layers, the model was damaged, so we set the maximum pruning rate in the paper to 16 layers.

## 6 OBTAINING THE BEST PRUNED MODELS

In Section 4 and Section 5, we have gained some valuable non-trivial practices and insights on LLM layer pruning through systematic experiments. Herein, we use these practices and insights to obtain the **Llama-3.1-6.3B-It** model and compare its performance against multiple baselines: (1) the original Llama-3.1-8B-It model, (2) a set of similarly sized community models and (3) a set of

Table 8: Performance of the Llama-3.1-6.3B-It models with respect to similarly-sized community models and state-of-the-art pruned models obtained through LLM layer pruning. All evaluations run by us. "Avg Acc" denotes the average accuracy calculated among eight datasets. "TTokens" denotes the training tokens. The best results are marked in **boldface**, and the sub-optimal ones are underlined.

| Baseline | # Parameters (TTokens) | Benchmarks | | | | | | | | Avg Acc |
|---|---|---|---|---|---|---|---|---|---|---|
| | | PIQA | HellaSwag | OpenbookQA | ARC-e | ARC-c | MMLU | CMMLU | WinoGrande | |
| Vicuna-7B-v1.5 | 6.74B (370M) | 0.7720±0.0098 | 0.5642±0.0049 | 0.3300±0.0210 | 0.7555±0.0088 | 0.4326±0.0145 | 0.4858±0.0040 | 0.3518±0.0044 | 0.6953±0.0129 | 0.5484 |
| ChatGLM2-6B | 6.24B (1.4T) | 0.5403±0.0116 | 0.2589±0.0044 | 0.1420±0.0156 | 0.2597±0.0090 | 0.2005±0.0117 | 0.2431±0.0036 | 0.2537±0.0040 | 0.5288±0.0140 | 0.3034 |
| Baichuan2-7B | 7.51B (2.6T) | 0.7666±0.0099 | 0.5363±0.0050 | 0.3020±0.0206 | 0.7475±0.0089 | 0.4206±0.0144 | 0.5024±0.0040 | 0.5220±0.0045 | 0.6819±0.0131 | 0.5599 |
| Qwen1.5-7B | 7.72B (18T) | 0.7845±0.0096 | 0.5785±0.0049 | 0.3160±0.0208 | 0.7125±0.0093 | 0.4053±0.0143 | 0.5967±0.0039 | **0.7277±0.0039** | 0.6575±0.0133 | 0.5973 |
| LLaMA3-8B | 8.03B (15T+) | 0.7965±0.0094 | 0.6014±0.0049 | **0.3480±0.0213** | 0.8005±0.0082 | 0.4983±0.0146 | 0.6212±0.0038 | 0.4752±0.0045 | 0.7332±0.0124 | 0.6093 |
| Gemma2-7B | 8.54B (6T) | **0.8025±0.0093** | **0.6039±0.0049** | 0.3300±0.0210 | 0.8110±0.0080 | 0.5009±0.0146 | 0.6143±0.0039 | 0.4430±0.0045 | **0.7435±0.0123** | 0.6061 |
| Llama-3.1-8B-It | 8.03B (15T+) | 0.8003±0.0093 | 0.5910±0.0049 | 0.3380±0.0212 | **0.8182±0.0079** | **0.5179±0.0146** | **0.6790±0.0038** | 0.5552±0.0045 | 0.7395±0.0123 | **0.6299** |
| ShortGPT (BI) | 6.29B (12.74M) | 0.7176±0.0105 | 0.4196±0.0049 | 0.2020±0.0180 | 0.6107±0.0100 | 0.2841±0.0132 | 0.2417±0.0036 | 0.2494±0.0040 | 0.5391±0.0140 | 0.4080 |
| Shortened LLaMA (PPL) | 6.29B (12.74M) | 0.7628±0.0099 | 0.4931±0.0050 | 0.2640±0.0197 | 0.7290±0.0091 | 0.3805±0.0142 | 0.3367±0.0040 | 0.2724±0.0041 | 0.5793±0.0139 | 0.4772 |
| Shortened LLaMA (Taylor) | 6.29B (12.74M) | 0.7138±0.0105 | 0.4964±0.0050 | 0.2740±0.0200 | 0.6848±0.0095 | 0.4181±0.0144 | 0.2861±0.0038 | 0.2504±0.0040 | 0.7135±0.0127 | 0.4796 |
| Llama-3.1-6.3B-It-Alpaca | 6.29B (12.74M) | 0.7383±0.0103 | 0.5323±0.0050 | 0.3080±0.0207 | 0.7260±0.0092 | 0.4684±0.0146 | 0.6567±0.0038 | 0.5515±0.0045 | 0.6646±0.0133 | 0.5807 |
| Llama-3.1-6.3B-It-Dolly | 6.29B (14.96M) | 0.7709±0.0098 | 0.5541±0.0050 | 0.3000±0.0205 | 0.7424±0.0090 | 0.4838±0.0146 | 0.6753±0.0038 | 0.5522±0.0045 | 0.7032±0.0128 | 0.5977 |

Table 9: The statistic of Llama-3.1-6.3B-It-Alpaca and Llama-3.1-6.3B-Dolly.

| Model | # Params | # MACs | Memory | Latency |
|---|---|---|---|---|
| Llama-3.1-6.3B-It-Alpaca, Llama-3.1-6.3B-Dolly | 6.29B | 368.65G | 23984MiB | 210.35s |

pruned models obtained by state-of-the-art LLM layer pruning methods (all prune 8 layers, fine-tune on Alpaca-cleaned).

Specifically, Llama-3.1-6.3B-It is obtained by pruning 8 layers of Llama-3.1-8B-It using the reverse-order metric. Note that, in contrast to these community models trained from scratch on trillions of tokens (except for Vicuna-7B-v1.5), Llama-3.1-6.3B-It is fine-tuned solely on Alpaca-cleaned (12.74M tokens) and Dolly-15k (14.96M tokens). For ease of distinction, we refer to them as "**Llama-3.1-6.3B-It-Alpaca**" and "**Llama-3.1-6.3B-It-Dolly**", respectively. From Table 8, we find that both Llama-3.1-6.3B-It-Alpaca and Llama-3.1-6.3B-It-Dolly outperform ChatGLM2-6B (GLM et al., 2024), Vicuna-7B-v1.5 (Zheng et al., 2024) and Baichuan2-7B (Baichuan, 2023), and partially exceed LLaMA3-8B (AI@Meta, 2024), Gemma2-7B (Team et al., 2024) (e.g., MMLU), while using significantly fewer training tokens. Notably, Llama-3.1-6.3B-It-Dolly also outperforms Qwen1.5-7B (Yang et al., 2024a). Besides, we also compare our models to other pruned models obtained by various LLM layer pruning methods. Experimental results show that our models are nearly 19% better than ShortGPT (Men et al., 2024) and 10%+ better than Shortened LLaMA (Kim et al., 2024). Table 9 presents the statistic of Llama-3.1-6.3B-It, including parameters, MACs, memory requirements and latency. Following Ma et al. (2023a), the statistical evaluation is conducted in inference mode, where the model is fed a sentence consisting of 64 tokens. The latency is tested under the test set of WikiText2 on a single NVIDIA RTX A100 GPU. We also present the generation results of the Llama-3.1-6.3B-It-Alpaca, Llama-3.1-6.3B-It-Dolly and Llama-3.1-8B-It in Table E.

## 7  CONCLUSION

In this paper, we revisit LLM layer pruning, focusing on pruning metrics, fine-tuning methods and pruning strategies. From these efforts, we have developed a practical list of best practices for LLM layer pruning. We use these practices and insights to guide the pruning of Llama-3.1-8B-Instruct and obtain Llama-3.1-6.3B-It-Alpaca and Llama-3.1-6.3B-It-Dolly. Our pruned models require fewer training tokens compared to training from scratch, yet still performing favorably against various popular community LLMs of similar size. We hope our work will help inform best practices for deploying LLMs in real-world applications.

**Limitations and Future Work.** In Section 5, we find that SFT datasets do effect the performance of pruned models. Therefore, we will explore which SFT datasets are more suitable for fine-tuning pruned models in future work. Additionally, in this paper, we focus primarily on layer pruning due to the straightforward nature of pruning layers in LLMs, where the input and output dimensions are identical. However, we plan to further investigate weight pruning (Sun et al., 2023; Frantar & Alistarh, 2023) and width pruning (Xia et al., 2023; Ma et al., 2023b) in future experiments.

## 8 REPRODUCIBILITY STATEMENT

The authors have made great efforts to ensure the reproducibility of the empirical results reported in this paper. Firstly, the experiment settings, evaluation metrics, and datasets were described in detail in Section 3.2. Secondly, the code to reproduce the results is available at `https://anonymous.4open.science/r/Navigation-LLM-layer-pruning-DEB7`, and the optimal model weights can be found at at `https://huggingface.co/anonymousICLR/Llama-3.1-6.3B-It-Alpaca` and `https://huggingface.co/anonymousICLR/Llama-3.1-6.3B-It-Dolly/`.

## 9 ETHICS STATEMENT

In this paper, we carefully consider ethical concerns related to our research and ensure that all methodologies and experimental designs adhere to ethical standards. Our study focuses on layer pruning to enhance the efficiency of LLMs and reduce computational resource requirements, thereby promoting sustainable AI development. Furthermore, all models and datasets used in our research are sourced from publicly available and accessible origins, ensuring no infringement on intellectual property or personal privacy.

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

# A SUPPLEMENTARY MATERIAL OF REASSESSING LAYER PRUNING IN LLMS: NEW INSIGHTS AND METHODS

Table A: Zero-shot performance of the pruned models (50% pruning rate, fine-tuning using LoRA). "Avg Acc" denotes the average accuracy calculated among eight datasets. The best results are marked in **boldface**, and the sub-optimal ones are underlined.

| Model | Metric | Benchmarks | | | | | | | | Avg Acc |
|---|---|---|---|---|---|---|---|---|---|---|
| | | PIQA | HellaSwag | OpenbookQA | ARC-e | ARC-c | MMLU | CMMLU | WinoGrande | |
| Vicuna-7B-v1.5 | Dense | 0.7720±0.0098 | 0.5642±0.0049 | 0.3300±0.0210 | 0.7555±0.0088 | 0.4326±0.0145 | 0.4858±0.0040 | 0.3518±0.0044 | 0.6953±0.0129 | 0.5484 |
| | Reverse-order | 0.5642±0.0116 | 0.2919±0.0045 | 0.1700±0.0168 | 0.3258±0.0096 | **0.2645±0.0129** | **0.4372±0.0041** | **0.3069±0.0043** | **0.5872±0.0138** | 0.3685 |
| | Random | 0.5773±0.0115 | 0.3083±0.0046 | 0.1560±0.0162 | 0.3775±0.0099 | 0.2176±0.0121 | 0.2650±0.0037 | 0.2542±0.0041 | 0.5067±0.0141 | 0.3328 |
| | PPL | **0.6572±0.0111** | **0.3524±0.0048** | **0.1940±0.0177** | **0.4971±0.0103** | 0.2406±0.0125 | 0.2361±0.0036 | 0.2510±0.0040 | 0.5328±0.0140 | **0.3702** |
| | Magnitude-l1 | 0.5239±0.0117 | 0.2585±0.0044 | 0.1400±0.0155 | 0.2635±0.0090 | 0.2184±0.0121 | 0.2295±0.0035 | 0.2527±0.0040 | 0.4893±0.0140 | 0.2970 |
| | Magnitude-l2 | 0.5245±0.0117 | 0.2590±0.0044 | 0.1300±0.0151 | 0.2656±0.0091 | 0.2210±0.0121 | 0.2293±0.0035 | 0.2512±0.0040 | 0.4791±0.0140 | 0.2950 |
| | BI | 0.5250±0.0117 | 0.2598±0.0044 | 0.1440±0.0157 | 0.2740±0.0092 | 0.1928±0.0115 | 0.2296±0.0035 | 0.2476±0.0040 | 0.4988±0.0141 | 0.2965 |
| | Taylor | 0.5283±0.0116 | 0.2585±0.0044 | 0.1300±0.0151 | 0.2572±0.0090 | 0.2167±0.0120 | 0.2614±0.0037 | 0.2513±0.0040 | 0.4901±0.0140 | 0.2992 |
| Qwen1.5-7B | Dense | 0.7845±0.0096 | 0.5785±0.0049 | 0.3160±0.0208 | 0.7125±0.0093 | f0.4053±0.0143 | 0.5967±0.0039 | 0.7277±0.0039 | 0.6575±0.0133 | 0.5973 |
| | Reverse-order | 0.5783±0.0115 | 0.3100±0.0046 | 0.1640±0.0166 | 0.3047±0.0094 | **0.2363±0.0124** | 0.2507±0.0037 | **0.2564±0.0041** | 0.5391±0.0140 | 0.3299 |
| | Random | 0.6409±0.0112 | **0.3268±0.0047** | **0.1940±0.0177** | **0.4617±0.0102** | 0.2261±0.0122 | 0.2321±0.0036 | 0.2529±0.0040 | 0.5083±0.0141 | **0.3553** |
| | PPL | **0.6529±0.0111** | 0.3233±0.0047 | 0.1700±0.0168 | 0.4360±0.0102 | 0.2099±0.0119 | 0.2297±0.0035 | 0.2541±0.0041 | 0.5225±0.0140 | 0.3498 |
| | Magnitude-l1 | 0.5452±0.0116 | 0.2690±0.0044 | 0.1280±0.0150 | 0.2837±0.0092 | 0.1962±0.0116 | 0.2548±0.0037 | 0.2479±0.0040 | 0.4862±0.0140 | 0.3013 |
| | Magnitude-l2 | 0.5348±0.0116 | 0.2651±0.0044 | 0.1520±0.0161 | 0.2858±0.0093 | 0.1843±0.0113 | **0.2659±0.0037** | 0.2519±0.0040 | 0.5059±0.0141 | 0.3057 |
| | BI | 0.6001±0.0114 | 0.2905±0.0045 | 0.1880±0.0175 | 0.4099±0.0101 | 0.2090±0.0103 | 0.2420±0.0036 | 0.2472±0.0040 | 0.4901±0.0140 | 0.3346 |
| | Taylor | 0.5223±0.0117 | 0.2540±0.0043 | 0.1460±0.0158 | 0.2403±0.0088 | 0.2176±0.0121 | 0.2393±0.0036 | 0.2478±0.0040 | 0.4854±0.0140 | 0.2941 |
| Gemma2-2B-It | Dense | 0.7867±0.0096 | 0.5367±0.0050 | 0.3560±0.0214 | 0.8085±0.0081 | 0.5111±0.0146 | 0.5687±0.0039 | 0.4499±0.0045 | 0.6961±0.0129 | 0.5892 |
| | Reverse-order | 0.6050±0.0114 | 0.3049±0.0046 | 0.1900±0.0176 | 0.3817±0.0100 | 0.2491±0.0126 | 0.2327±0.0036 | 0.2527±0.0040 | 0.5580±0.0140 | 0.3468 |
| | Random | **0.6741±0.0109** | 0.3441±0.0047 | 0.2180±0.0185 | 0.5446±0.0102 | 0.2696±0.0130 | 0.2307±0.0036 | **0.2540±0.0041** | 0.5335±0.0140 | 0.3836 |
| | PPL | 0.6621±0.0110 | **0.3505±0.0048** | **0.2380±0.0191** | 0.5585±0.0102 | 0.2526±0.0127 | 0.2328±0.0036 | 0.2526±0.0040 | 0.5280±0.0140 | **0.3844** |
| | Magnitude-l1 | 0.6649±0.0110 | 0.3358±0.0047 | 0.1960±0.0178 | **0.5564±0.0102** | 0.2355±0.0124 | 0.2307±0.0035 | 0.2516±0.0040 | 0.5264±0.0140 | 0.3747 |
| | Magnitude-l2 | 0.6159±0.0113 | 0.2956±0.0046 | 0.1720±0.0169 | 0.4301±0.0102 | 0.2073±0.0118 | 0.2319±0.0036 | 0.2501±0.0040 | 0.5178±0.0140 | 0.3401 |
| | BI | 0.6376±0.0111 | 0.3310±0.0047 | 0.2140±0.0184 | 0.4891±0.0103 | 0.2406±0.0125 | **0.2397±0.0036** | 0.2532±0.0040 | 0.5667±0.0139 | 0.3715 |
| | Taylor | 0.6088±0.0114 | 0.3142±0.0046 | 0.1880±0.0175 | 0.4049±0.0101 | **0.2739±0.0130** | 0.2297±0.0035 | 0.2508±0.0040 | **0.5817±0.0139** | 0.3565 |
| Llama-3.1-8B-It | Dense | 0.8003±0.0093 | 0.5910±0.0049 | 0.3380±0.0212 | 0.8182±0.0079 | 0.5179±0.0146 | 0.6790±0.0038 | 0.5552±0.0045 | 0.7395±0.0123 | 0.6299 |
| | Reverse-order | 0.6376±0.0112 | 0.3163±0.0046 | **0.1960±0.0178** | 0.4019±0.0101 | **0.3106±0.0135** | **0.2502±0.0036** | 0.2482±0.0040 | **0.6101±0.0137** | **0.3714** |
| | Random | 0.5588±0.0116 | 0.2730±0.0044 | 0.1280±0.0150 | 0.2826±0.0093 | 0.1903±0.0115 | 0.2406±0.0036 | **0.2555±0.0041** | 0.5020±0.0141 | 0.3039 |
| | PPL | **0.6643±0.0110** | **0.3548±0.0048** | **0.1960±0.0178** | **0.4718±0.0102** | 0.2483±0.0126 | 0.2394±0.0036 | 0.2446±0.0040 | 0.5454±0.0140 | 0.3706 |
| | Magnitude-l1 | 0.5316±0.0116 | 0.2576±0.0044 | 0.1360±0.0153 | 0.2572±0.0090 | 0.1980±0.0116 | 0.2344±0.0036 | 0.2526±0.0040 | 0.4933±0.0141 | 0.2951 |
| | Magnitude-l2 | 0.5316±0.0116 | 0.2576±0.0044 | 0.1360±0.0153 | 0.2572±0.0090 | 0.1980±0.0116 | 0.2344±0.0036 | 0.2526±0.0040 | 0.4933±0.0141 | 0.2951 |
| | BI | 0.5773±0.0115 | 0.2878±0.0045 | 0.1520±0.0161 | 0.3674±0.0099 | 0.1706±0.0110 | 0.2342±0.0036 | 0.2466±0.0040 | 0.5036±0.0141 | 0.3174 |
| | Taylor | 0.6088±0.0114 | 0.3288±0.0047 | 0.1660±0.0167 | 0.4318±0.0102 | 0.2790±0.0131 | 0.2310±0.0036 | 0.2534±0.0041 | 0.6093±0.0137 | 0.3635 |

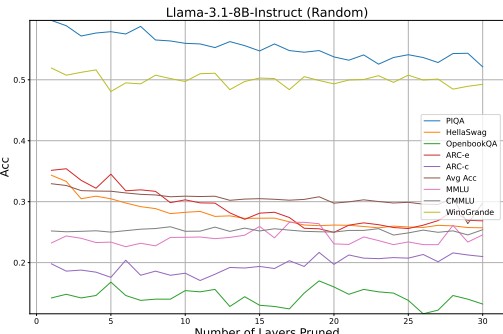

Figure A: The effect of different pruning rates on LLM layer pruning using random metric.

Table B: Zero-shot performance of the pruned models using various fine-tuning methods under 25% pruning rate (using taylor metric). "Avg Acc" denotes the average accuracy calculated among eight datasets. The best results are marked in **boldface**, and the sub-optimal ones are underlined.

| Model | Method | Layer | Benchmarks | | | | | | | | Avg Acc |
|---|---|---|---|---|---|---|---|---|---|---|---|
| | | | PIQA | HellaSwag | OpenbookQA | ARC-e | ARC-c | MMLU | CMMLU | WinoGrande | |
| Llama-3.1-8B-It | LoRA | - | 0.7138±0.0105 | 0.4964±0.0050 | 0.2740±0.0200 | 0.6848±0.0095 | 0.4181±0.0144 | 0.2861±0.0038 | 0.2504±0.0040 | 0.7135±0.0127 | 0.4796 |
| | QLoRA | - | 0.6496±0.0111 | 0.3260±0.0047 | 0.1820±0.0173 | 0.4520±0.0102 | 0.2969±0.0134 | 0.3425±0.0040 | 0.2627±0.0041 | 0.5793±0.0139 | 0.3864 |
| | | *lm_head only* | 0.6752±0.0109 | 0.3685±0.0048 | 0.2100±0.0182 | 0.5349±0.0102 | 0.3276±0.0137 | 0.4315±0.0041 | 0.3373±0.0044 | 0.6795±0.0109 | 0.4456 |
| | Partial-layer | *lm_head+last layer* | 0.7029±0.0107 | 0.4676±0.0050 | 0.2140±0.0184 | 0.6393±0.0099 | 0.3763±0.0142 | 0.5682±0.0041 | 0.4483±0.0046 | 0.6748±0.0132 | 0.5114 |
| | | *lm_head+last two layers* | 0.7252±0.0104 | 0.5173±0.0050 | 0.2800±0.0201 | 0.7104±0.0093 | 0.4232±0.0144 | 0.6058±0.0040 | 0.4659±0.0046 | 0.7040±0.0128 | 0.5540 |
| | | *lm_head+last three layers* | **0.7345±0.0103** | **0.5290±0.0050** | **0.3020±0.0206** | **0.7399±0.0090** | **0.4360±0.0145** | **0.6277±0.0039** | **0.4763±0.0046** | **0.7151±0.0127** | **0.5701** |

Table C: Zero-shot performance of pruned models (50% pruning rate) using different pruning strategies. "Avg Acc" denotes the average accuracy calculated among eight datasets. The best results are marked in **boldface**. "1:1:12" refers to an iterative pruning process where 1 layer is pruned at a time, and a total of 12 layers are pruned by the end of the process.

| Fine-tuning Method | Model | Method | Iteration steps | Benchmarks | | | | | | | | Avg Acc |
|---|---|---|---|---|---|---|---|---|---|---|---|---|
| | | | | PIQA | HellaSwag | OpenbookQA | ARC-e | ARC-c | MMLU | CMMLU | WinoGrande | |
| LoRA | Llama-3.1-8B-It | Reverse-order | one-shot | 0.6376±0.0112 | 0.3163±0.0046 | 0.1960±0.0178 | 0.4019±0.0101 | 0.3106±0.0135 | 0.2502±0.0036 | 0.2482±0.0040 | 0.6101±0.0137 | 0.3714 |
| | | | 1:8:16 | 0.6376±0.0112 | 0.3160±0.0046 | 0.1980±0.0178 | 0.3990±0.0100 | 0.3106±0.0135 | 0.2526±0.0037 | 0.2504±0.0040 | 0.6046±0.0137 | 0.3711 |
| | | | 1:1:16 | 0.6333±0.0112 | 0.3259±0.0047 | 0.2020±0.0180 | 0.4146±0.0101 | 0.2961±0.0133 | 0.2426±0.0036 | 0.2690±0.0041 | 0.5912±0.0138 | **0.3718** |
| | | Taylor | one-shot | 0.6088±0.0114 | 0.3288±0.0047 | 0.1660±0.0167 | 0.4318±0.0102 | 0.2790±0.0131 | 0.2310±0.0036 | 0.2534±0.0041 | 0.6093±0.0137 | **0.3635** |
| | | | 1:8:16 | 0.6230±0.0113 | 0.3516±0.0048 | 0.1480±0.0159 | 0.4604±0.0102 | 0.2355±0.0124 | 0.2541±0.0037 | 0.2546±0.0041 | 0.5312±0.0140 | 0.3573 |
| | | | 1:1:16 | 0.5430±0.0113 | 0.2692±0.0044 | 0.1580±0.0163 | 0.2921±0.0093 | 0.1937±0.0115 | 0.2334±0.0036 | 0.2481±0.0040 | 0.5091±0.0141 | 0.3058 |
| | Gemma2-2B-It | Reverse-order | one-shot | 0.6050±0.0114 | 0.3049±0.0046 | 0.1900±0.0176 | 0.3817±0.0100 | 0.2491±0.0126 | 0.2327±0.0036 | 0.2527±0.0040 | 0.5580±0.0140 | 0.3468 |
| | | | 1:6:12 | 0.6007±0.0114 | 0.3076±0.0046 | 0.1900±0.0176 | 0.3994±0.0101 | 0.2483±0.0126 | 0.2429±0.0036 | 0.2495±0.0040 | 0.5478±0.0140 | **0.3483** |
| | | | 1:1:12 | 0.6023±0.0114 | 0.3173±0.0046 | 0.1720±0.0169 | 0.3897±0.0100 | 0.2449±0.0126 | 0.2531±0.0037 | 0.2481±0.0040 | 0.5387±0.0140 | 0.3458 |
| | | Taylor | one-shot | 0.6088±0.0114 | 0.3142±0.0046 | 0.1880±0.0175 | 0.4049±0.0101 | 0.2739±0.0130 | 0.2297±0.0035 | 0.2508±0.0040 | 0.5817±0.0139 | 0.3565 |
| | | | 1:6:12 | 0.5909±0.0115 | 0.2806±0.0045 | 0.1380±0.0154 | 0.3834±0.0100 | 0.2150±0.0120 | 0.2295±0.0035 | 0.2523±0.0040 | 0.5059±0.0141 | 0.3245 |
| | | | 1:1:12 | 0.6502±0.0111 | 0.3456±0.0047 | 0.1860±0.0174 | 0.4790±0.0103 | 0.2483±0.0126 | 0.2314±0.0036 | 0.2578±0.0041 | 0.5525±0.0140 | **0.3689** |
| Partial-layer | Llama-3.1-8B-It | Reverse-order | one-shot | 0.6578±0.0111 | 0.4137±0.0049 | 0.2200±0.0185 | 0.5707±0.0102 | 0.3294±0.0137 | 0.3854±0.0040 | 0.3190±0.0043 | 0.6504±0.0134 | 0.4433 |
| | | | 1:1:16 | 0.6774±0.0109 | 0.4164±0.0049 | 0.2200±0.0185 | 0.5863±0.0101 | 0.3362±0.0138 | 0.4170±0.0041 | 0.3460±0.0044 | 0.6385±0.0135 | **0.4547** |
| | | Taylor | one-shot | 0.6649±0.0110 | 0.3985±0.0049 | 0.2100±0.0182 | 0.5581±0.0102 | 0.3251±0.0137 | 0.3054±0.0039 | 0.2876±0.0042 | 0.6212±0.0136 | 0.4214 |
| | | | 1:1:16 | 0.5876±0.0115 | 0.2813±0.0045 | 0.1300±0.0151 | 0.3986±0.0100 | 0.1980±0.0116 | 0.2508±0.0037 | 0.2502±0.0040 | 0.4957±0.0141 | 0.3240 |

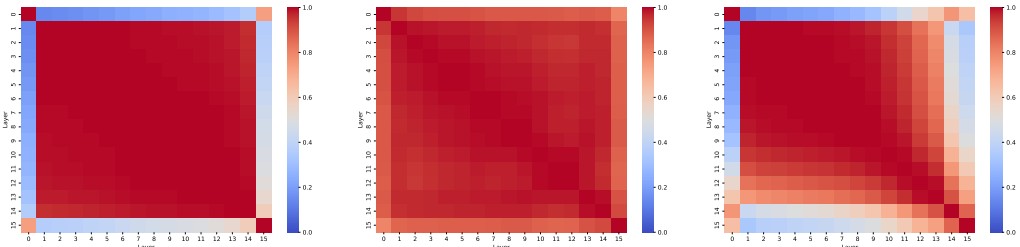

Figure B: Visualization of the layer similarity matrix of 16-layer Llama-3.1-8B-It models (using Taylor) obtained by different pruning strategies. Left: one-shot pruning; Middle: iterative pruning with pruning step = 1; Right: iterative pruning with pruning step = 8.

Table D: The effect of number of calibration samples on LLM layer pruning. Detailed version of Table 4.

| Model | Metric | Calibration Samples | Removed Layers | Benchmarks | | | | | | | | Avg Acc |
|---|---|---|---|---|---|---|---|---|---|---|---|---|
| | | | | PIQA | HellaSwag | OpenbookQA | ARC-e | ARC-c | MMLU | CMMLU | WinoGrande | |
| Llama-3.1-8B-Instruct | BI | 1 | 2,3,5,6,7,8,11,12 | 0.7029±0.0107 | 0.4167±0.0049 | 0.2060±0.0181 | 0.6136±0.0100 | 0.2739±0.0130 | 0.2362±0.0036 | 0.2512±0.0040 | 0.5225±0.0140 | 0.40 |
| | | 5 | 3,4,5,8,9,10,13,19 | 0.7236±0.0104 | 0.4400±0.0050 | 0.2420±0.0192 | 0.6730±0.0096 | 0.3311±0.0138 | 0.2524±0.0037 | 0.2553±0.0041 | 0.5485±0.0140 | 0.43 |
| | | 10 | 2,3,4,5,6,7,8,9 | 0.7176±0.0105 | 0.4196±0.0049 | 0.2020±0.0180 | 0.6107±0.0100 | 0.2841±0.0132 | 0.2417±0.0036 | 0.2494±0.0040 | 0.5391±0.0140 | 0.41 |
| | | 30 | 2,3,4,10,11,12,13,14 | 0.7209±0.0105 | 0.4328±0.0049 | 0.2040±0.0180 | 0.6414±0.0098 | 0.3259±0.0137 | 0.2500±0.0036 | 0.2576±0.0041 | 0.5517±0.0140 | 0.42 |
| | | 50 | 2,3,4,5,6,7,10,13 | 0.7100±0.0106 | 0.4091±0.0049 | 0.2180±0.0185 | 0.6221±0.0099 | 0.2875±0.0132 | 0.2492±0.0036 | 0.2529±0.0040 | 0.5462±0.0140 | 0.41 |
| | Taylor | 1 | 27, 26, 25, 24, 28, 23, 29, 22 | 0.6088±0.0114 | 0.3288±0.0047 | 0.1660±0.0167 | 0.4318±0.0102 | 0.2790±0.0131 | 0.2310±0.0036 | 0.2534±0.0041 | 0.6093±0.0137 | 0.36 |
| | | 5 | 24, 26, 25, 28, 27, 23, 29, 22 | 0.6088±0.0114 | 0.3288±0.0047 | 0.1660±0.0167 | 0.4318±0.0102 | 0.2790±0.0131 | 0.2310±0.0036 | 0.2534±0.0041 | 0.6093±0.0137 | 0.36 |
| | | 10 | 24, 26, 25, 28, 27, 23, 29, 22 | 0.6088±0.0114 | 0.3288±0.0047 | 0.1660±0.0167 | 0.4318±0.0102 | 0.2790±0.0131 | 0.2310±0.0036 | 0.2534±0.0041 | 0.6093±0.0137 | 0.36 |
| | | 30 | 24, 23, 25, 26, 22, 27, 28, 20 | 0.7280±0.0104 | 0.4985±0.0050 | 0.2460±0.0193 | 0.6961±0.0094 | 0.4130±0.0144 | 0.6611±0.0038 | 0.4915±0.0046 | 0.7032±0.0128 | 0.55 |
| | | 50 | 24, 23, 25, 26, 22, 27, 28, 20 | 0.7280±0.0104 | 0.4985±0.0050 | 0.2460±0.0193 | 0.6961±0.0094 | 0.4130±0.0144 | 0.6611±0.0038 | 0.4915±0.0046 | 0.7032±0.0128 | 0.55 |

Table E: Generated Examples from the Llama-3.1-6.3B-It-Alpaca, Llama-3.1-6.3B-It-Dolly and Llama-3.1-8B-It.

| Model | Generated Sentences |
|---|---|
| Llama-3.1-6.3B-It-Alpaca | To be or not to be, that is the question that has been asked thousands of times throughout history. but have you ever wondered what it would be like to not exist? what would it feel like to not have consciousness or awareness of any kind? would it be like being asleep or a combed out hair? no, it would be something else entirely. it would be like not being here at all, like being a tree or a rock, a piece of furniture or a piece of trash. nothingness is not something we can really comprehend because we can't experience it. we can't even imagine it because we can't imagine. |
| | It fills me with such pride and joy knowing that I have been able to help shaped the lives of countless students over the years. I am proud of the numerous accomplishments and achievements of my students, and I am proud of the role I have played in shaping them into the people they are today. As a teacher, I believe it is my duty to provide an optimal learning environment that fosters growth and development of my students. I believe in making learning fun and engaging, and I am dedicated to pushing the bounds of innovation in the classroom. I am dedicated to ensuring that my students leave my classroom with the knowledge and skills they need ... |
| | What's great about the holiday season? Christmas lights, Christmas shows, Christmas presents, Christmas holiday traditions. But what's not great about the holiday season? crowds, stress, Santa Claus, Christmas holiday stress, Christmas holiday stressors. It's important to remember to do things that help you relax during the holiday season, such as taking time for yourself, engaging in relaxation techniques, practicing mindfulness, engaging in physical activity, practicing gratitude, practicing self-care, engaging in activities that bring you joy, and spending time with loved ones. These are all important components of stressors prevention during the holiday season. Here are some tips to help you. ... |
| Llama-3.1-6.3B-It-Dolly | To be or not to be, that is the question - Hamlet's famous soliloquy - could easily apply to my current state of mind. I've been struggling to decision whether to go back to school or not. On one hand, I'm really interesting in learning more about the subject matter I've been passionate about for years, and I think pursuing a degree would be a great way to formalize my knowledge. On the Hamlet scale, this option would be "to be": me myself, fully engaged in learning and growing as a person. On the other hand, I've been married for almost 20 years ... |
| | It fills me with such pride and joy to see my little human growing up. yesterday was a special day in our household because we celebrated one of the major milestone in life. my little one cut his first teeth. Yes you heard it right he is now a big boy and his first teeth are now visible. It has been a long journey since we started noticed his teething symptoms and we have been through lot of sleepless nights and teething tantrums but it all pay off now. I can see the relief in his little face when he finally gets to experience the world without the constant pain in his little bottom teeth. As a ... |
| | What's great about the holiday season is spending time with loved ones and making new traditions. But let's not forget about the stress and pressure that comes with it. We're here to help you beat stress during the holiday season with some tips from our #Ask a therapist series with licensed therapist, Dr. Dezympfer. ### tip #1: Prioritize Self-Cark. With all the commotion during the holiday season, it can be easy to forget to take care of yourself. Make sure to schedule time for self-Cark. whether that is taking a yoga class, taking a few minutes each day to meditate, ... |
| Llama-3.1-8B-It | To be or not to be, that is the question. How does one decide whether a being is alive or not? The issue of what constitutes life has puzzled philosophers and scientists alike for centuries. A being is alive if it meets the criteria of life, which are: Metabolism: The ability to carry out basic functions such as respiration, photosynthesis, and nutrient uptake. Homeostasis: The ability to maintain a stable internal environment, despite changes in the external environment. Growth and Development: The ability to increase in size and complexity over time. Reproduction: The ability to produce offspring, either sexually or asex |
| | It fills me with such pride and joy to see my friends' passions ignite into flame and to watch them soar to great heights! It was an absolutely honour to have been able to share this experience with you and to bear witness to your hard work and dedication! It was an absolute pleasure to get to know you and share our experiences, but I know that this is just the beginning of an amazing journey! I feel so grateful to have you in my life, and I am constantly in awe of your love, kindness, and generosity! It's been such an incredible journey watching you discover and grow, and I feel so lucky to have... |
| | What's great about the holiday season? In addition to being able to spend time with loved ones and get some much-needed R&R, many people enjoy the idea of giving back to others. Whether it's volunteering, donating to charity, or participating in a Secret Santa gift exchange, the holiday season can be a time of kindness and generosity. But have you ever thought about how you might be able to combine your love of cooking and giving back this holiday season? If so, you might be interested in hosting a charity-themed potluck dinner or bake sale. Here are a few ideas to get you started: Host a potluck dinner to... |

