# OpenReview forum: "Reassessing Layer Pruning in LLMs: New Insights and Methods"
_ICLR.cc/2025/Conference — ICLR 2025 Conference Withdrawn Submission_

### Official Review · Reviewer_6JxQ · 2024-10-31

**Soundness:** 2
**Presentation:** 2
**Contribution:** 2
**Rating:** 3
**Confidence:** 5

**Summary:**

This paper mainly focuses on the ablation study of layer pruning in LLMs.
The paper first explores the different layer pruning strategies with different fine-tuning methods.
Then, they find that the reverse-order is the optimal layer pruning strategy.
Meanwhile. they find that the partial-layer fine-tuning outperforms LoRA-based techniques.
Finally, they release two models directly pruned from Llama-3.1-8B-Instruct, which outperforms other popular models with similar sizes.

**Strengths:**

1. The ablation study about layer pruning and fine-tuning in this paper seems to be good.
2. The paper finds that the partial-layer fine-tuning outperforms LoRA-based techniques, which is important to the post-pruning fine-tuning research areas.

**Weaknesses:**

1. The novelty of this paper is limited. Most works have done in this paper are kind of ablation study. The paper does not propose any new method, the paper only provides the findings after the ablation study with its comprehensive benchmarking. The method of pruning layers in 'Reverse-order' is only the findings obtained from the ablation study compared to other methods, which is not a novel method. Meanwhile, the ablation of layer pruning methods is only conducted with models with around 7B or less parameters, which shows limited generalization to larger models.
2. The ablation study for layer pruning in Table 1 2 5 8 does not include the large models, for example LLaMA-2 30B, LLaMA-2 70B and LLaMA-3 80B, thus the generalization of this method on larger models is limited. And so does the ablation of fine-tuning. As the model becomes larger, the redundancy of the model becomes larger, which is more important to show the pruning results with large models, especially 70B or 80B models.
3. According to Table 7, it shows that the fine-tuning dataset is sensitive to the model performance, which hurts the generalization of this method. The work does not discuss the calibration dataset used for those other pruning methods, which results in the bias of the results. Meanwhile, the paper does not include the ablation study with different number of samples used in sft.

**Questions:**

1. How about the performance of this method when applied to large LLMs including LLaMA-2 30B, LLaMA-2 70B and LLaMA-3 80B? As it is intuitive to apply pruning techniques (especially layer pruning methods) on larger models (especially 70B or 80B) models, because there are much more redundancy compared to 7B model family.
2. How about the generation speed compared to other models with similar model size included in Table 8?
3. How about the ablation study with different number of training samples in sft?
4. What is the experiment setup for other layer pruning methods? especially, what is the number of samples for the calibration?

---

> ### Author Response · Authors · 2024-11-25
> **Response to Reviewer 6JxQ**
>
> **Most works have done in this paper are kind of ablation study.** We appreciate the reviewer's feedback regarding the novelty of our work. While we respectfully disagree with the characterization of our work as "ablation studies," as our research makes two significant contributions to the field:
>
> * Through systematic exploration, we discovered that 'Reverse-order' pruning consistently outperforms more complex pruning methods across diverse datasets and models. This finding is particularly impactful as it establishes a strong, reproducible baseline for LLM layer pruning - demonstrating that simpler approaches can be more effective than complicated pruning strategies. This challenges the common assumption that more sophisticated pruning methods are necessarily better.
>
> * Our second key finding - that fine-tuning only the last few remaining layers and lm_head outperforms popular methods like LoRA - represents a paradigm shift in LLM pruning optimization. This discovery provides practical benefits for efficient model adaptation.
>
> Our work is, to the best of our knowledge, the first to demonstrate these phenomena in LLM pruning, offering significant practical value through both the simple yet effective reverse-order pruning strategy and the empirical insights into the effectiveness of partial fine-tuning. These findings provide clear, actionable guidelines for practitioners working on LLM pruning while challenging common assumptions about the necessity of complex pruning methods and the optimality of LoRA-based approaches.
>
> The simplicity of our proposed methods is actually a strength, as it enables broader adoption and reproducibility while achieving superior results compared to more complex approaches. We believe these contributions advance the field's understanding of LLM pruning and provide practical tools for researchers and practitioners.
>
> **Do not include the large models.** To address this, we have conducted experiments on the LLaMA-3 70B model. However, due to limited time and computational resources, we are unable to perform fine-tuning experiments on this model and have focused instead on evaluating the performance of the pruned models without fine-tuning. In order to verify the effectiveness of the reverse-order pruning, we compare it with an advanced training-free layer pruning method SLEB [1] (mentioned by Reviewer uw4p). We prune 16 layers with these two methods. As shown in the table, we find that reverse-order pruning and SLEB have similar average performance, which demonstrates the effectiveness of reverse-order pruning. It is important to note that SLEB requires evaluating the importance of each layer, removing the least important layer from the current model, and then iterating through this verification process. This iterative approach of SLBE is time-consuming and takes up a lot of memory for LLaMA-3 70B. In contrast, reverse-order pruning is simple but effective.
>
> | Method | PIQA | HellaSwag | OpenbookQA | ARC-e | ARC-c | MMLU | CMMLU | WinoGrande | Avg Acc |
> |:---------:|:--------------:|:--------------:|:--------------:|:-------------:|:-------------:|:-------------:|:-------------:|:-------------:|:-------:|
>  | SLEB | 0.7916±0.0095 | 0.5805±0.0049 | 0.3360±0.0211 | 0.8005±0.0082 | 0.4974±0.0146 | 0.5604±0.0040 | 0.4125±0.0045 | 0.7238±0.0126 | 0.5878 |
> | Reverse-order | 0.7231±0.0104 | 0.4461±0.0050 | 0.3280±0.0210 | 0.6561±0.0097 | 0.4616±0.0146 | 0.7473±0.0034 | 0.6057±0.0044 | 0.6961±0.0129 | 0.5830 |
>  | Original | 0.8243±0.0089 | 0.6636±0.0047 | 0.3800±0.0217 | 0.8683±0.0069 | 0.6032±0.0143 | 0.7519±0.0033 | 0.6667±0.0042 | 0.8058±0.0111 | 0.6958 |
>
> **The generation speed compared to other models with similar model size included in Table 8.** To address this, we have measured the generation speed of other models with similar model size. As mentioned in line 522-524, the latency is tested under the test set of WikiText2 on a single NVIDIA RTX A100GPU. As shown in the table below, our pruned models are significantly faster than existing models with similar model size.
> | Model                          | Latency  |
> |:-------------------------------:|:--------:|
> | Llama-3.1-6.3B-It-Alpaca,Llama-3.1-6.3B-Dolly       | 210.35s  |
> | Vicuna-7B                       | 286.42s  |
> | Qwen1.5-7B                       | 270.48s  |
> | LLAMA3-8B                        | 277.24s  |
> | Baichuan2-7B                     | 324.99s  |
> | Llama-3.1-8B-Instruct           | 311.40s   |

---

> > ### Author Response · Authors · 2024-11-25
> > **Response to Reviewer 6JxQ (2)**
> >
> > **The ablation study with a different number of samples used in sft.** To answer your question, we conduct further experiments on the number of samples used in SFT. Specifically, we use 20%, 40%, 60%, 80% and 100% of the alpaca-cleaned dataset for partial fine-tuning. As shown in the table below, we find that the number of samples used in SFT indeed affects the performance of the pruned model. When there is only 20% of the dataset, the performance of the model declines significantly.
> >
> > | Data Quantity | PIQA           | HellaSwag      | OpenbookQA     | ARC-e         | ARC-c         | MMLU          | CMMLU         | WinoGrande    | Avg Acc |
> > |:------:|:--------------:|:--------------:|:--------------:|:-------------:|:-------------:|:-------------:|:-------------:|:-------------:|:-------:|
> > | 100%      | 0.7383±0.0103  | 0.5323±0.0050  | 0.3080±0.0207  | 0.7260±0.0092 | 0.4684±0.0146 | 0.6567±0.0038 | 0.5515±0.0045 | 0.6646±0.0133 | 0.5807  |
> > | 80%    | 0.7372±0.0103  | 0.5279±0.0050  | 0.3100±0.0207  | 0.7235±0.0092 | 0.4565±0.0146 | 0.6515±0.0038 | 0.5477±0.0045 | 0.6567±0.0133 |    0.5764     |
> > | 60%    | 0.7399±0.0102  | 0.5242±0.0050  | 0.3140±0.0208  | 0.7100±0.0093 | 0.4497±0.0145 | 0.6551±0.0038 | 0.5487±0.0045 | 0.6582±0.0133 |   0.5747      |
> > | 40%    | 0.7399±0.0102  | 0.5194±0.0050  | 0.3060±0.0206  | 0.7020±0.0094 | 0.4548±0.0146 | 0.6540±0.0038 | 0.5531±0.0045 | 0.6630±0.0133 |   0.5740      |
> > | 20%    | 0.7383±0.0103  | 0.5077±0.0050  | 0.2980±0.0205  | 0.6860±0.0095 | 0.4360±0.0145 | 0.6455±0.0038 | 0.5458±0.0045 | 0.6590±0.0133 |  0.5083       |
> >
> > **What is the experiment setup for other layer pruning methods.**  As we mentioned in line 207 and 208 of our initial submission, we utilize the Alpaca-cleaned dataset with LoRA to recover the performance for other layer pruning methods. As mentioned in lines 259 and 260, we use LoRA with a rank d of 8 and a batch size of 64, and the AdamW optimizer. The learning rate is set to 1×10−5 with 100 warming steps. We followed the guidelines of [2] and divided 2000 samples from the Alpaca dataset into a validation set, with the remaining samples used for training.
> >
> > To ensure fairness, the training sets are kept consistent across all pruning methods. In Table 8, LLaMA-3.1-6.3B-It-Alpaca is obtained by pruning 8 layers from LLaMA-3.1-8B-It using the Reverse Order metric, fine-tuning only the last three layers and the lm_head on the Alpaca-cleaned dataset, while freezing the remaining layers. ShortGPT (BI), Shortened LLaMA (PPL) and Shortened LLaMA (Taylor) are fine-tuned on the same dataset with LoRA. As shown in the Table 8 of our initial submission, Llama-3.1-6.3B-It-Alpaca are nearly 18% better than ShortGPT and 10%+ better than Shortened LLaMA, which demonstrates the effectiveness of our method.
> >
> > [1] Song, Jiwon, et al. "SLEB: Streamlining LLMs through Redundancy Verification and Elimination of Transformer Blocks." arXiv preprint arXiv:2402.09025 (2024).
> >
> > [2] Ma, Xinyin, Gongfan Fang, and Xinchao Wang. "Llm-pruner: On the structural pruning of large language models." Advances in neural information processing systems 36 (2023): 21702-21720.

---

> ### Comment · Reviewer_6JxQ · 2024-11-25
>
> Hi, thanks for your rebuttal and the new experiments for large model (LLaMA-3 70B) and the ablation for number of sft samples.
>
> The main concern of this paper still remains in the novelty and the contribution. Authors try to compare the reserve pruning with other pruning methods to show the effectiveness of the reserve pruning.
>
> However, the work [1] shows the method of structured pruning globally, and the work [2] also shows the global optimal small dense results. Thus, I am curious about that if the reserve layer pruning can achieve the global optimal pruning results? As there might be some redundancy in the front layers as shown by Figure 3 in work [1] and Figure 3 in work [3]. Could you explain for the global optimum or compare with the works [1] [2]?
>
> Meanwhile, works [1] [2] [4] [5] do not include the fine-tuning (backward) for further performance improvement. While this work adopts Alpaca dataset for the fine-tune, which shows limited generalization (i.e., rely on fine-tune dataset for performance recover). Other works [1] [2] [4] can recover with only WikiText2 dataset and work [5] even does not require calibration. Even the search effort and calibration effort of work [2] remain to the similar as the work [3], thus I think adopt Alpaca for fine-tuning is not that kind of necessary. In my opinion, it is a better way for us to adopt the non-backward method for LLMs to recover model performance as it can be generalized to the extra large models to save the resources. By the way, the works [1] [2] [4] [5] can all generate a smaller model, and are compatible to the further fine-tuning, and I may believe that these works may achieve comparable results as this work after fine-tuning.
>
> [1] Fluctuation-based Adaptive Structured Pruning for Large Language Models
>
> [2] Search for Efficient Large Language Models
>
> [3] LLM-Pruner: On the Structural Pruning of Large Language Models
>
> [4] SliceGPT: Compress Large Language Models by Deleting Rows and Columns
>
> [5] A Simple and Effective Pruning Approach for Large Language Models
>
> I am really respect to the authors' contribution to this paper and the rebuttal, but my concern to this paper's novelty still remains. Therefore, currently, I may retain my score.
>
> I am ready for further discussion.

---

> > ### Author Response · Authors · 2024-11-26
> > **Response to Reviewer 6JxQ (3)**
> >
> > We sincerely appreciate the reviewer’s thoughtful feedback and the opportunity to further clarify the contributions and positioning of our work. Below, we address the concerns regarding novelty and methodology, and we hope to provide additional insights for discussion.
> >
> > **1. Different angle and scope of our contribution.**
> >
> > While we acknowledge the effectiveness of global search methods for optimizing LLM structures, as highlighted in works [1] and [2], our paper approaches the problem from a different angle. Specifically, we focus on simplicity and practicality by proposing reverse-order pruning, which prioritizes ease of implementation and computational efficiency over theoretical guarantees of optimality.
> >
> > We intentionally did not claim global optimality for reverse-order pruning or any resulting model structure, as we recognize that such guarantees would require a rigorous theoretical framework beyond the scope of this paper, particularly given the complexity of LLMs. Instead, we emphasize in Insight #1 on page 2 that reverse-order pruning is a **simple yet effective** method. Its effectiveness has been validated through robust numerical results presented in the main paper and further substantiated during the rebuttal phase with experiments on LLaMA-3 70B and the ablation study on the number of SFT samples.
> >
> > **2. Focus on layer pruning rather than channel pruning or weight pruning.**
> >
> > We first give the definitions of layer pruning, channel pruning, and weight pruning.
> > * Layer pruning removes entire layers from the model, resulting in a simpler and coarser-grained reduction.
> > * Channel pruning removes specific channels (or neurons) from individual layers, resulting in a finer-grained reduction in the model's width.
> > * Weight pruning removes individual weights (connections) within layers. It is the most granular form of pruning, targeting the least important connections.
> >
> > Our work is primarily concerned with layer pruning, driven by the motivation to develop a coarse-grained yet effective approach. It is worth noting that the reverse-order pruning method can complete the pruning without additional information. In contrast, the cited works utilize **channel pruning** [1,2,3] and **weight pruning** [4,5], which typically requires fine-grained adjustments and additional computational effort to prune at a more granular level (e.g., weights or channels).
> >
> > We find that comparing layer pruning directly to channel pruning or weight pruning methods is not entirely fair, as they address different pruning objectives. Layer pruning offers the advantage of simplicity and computational efficiency. This practical focus aligns with our goal of presenting an easy-to-implement method that can be readily applied in real-world scenarios.
> >
> > **3. There might be some redundancy in the front layers.**
> >
> > The metric used in Figure 3 of work [1] to identify redundancy in the front layers is based on magnitude, which is also a baseline metric we evaluate in our paper (in line 176 of page 4). As shown in Table 1 of our initial submission, we demonstrate that the Reverse-order pruning outperforms magnitude-based approaches (both Magnitude-l1 and Magnitude-l2). This indicates that while [1] identifies redundancy using magnitude, our proposed method provides a more effective and reliable metric for achieving better performance after pruning.
> >
> > [1] Fluctuation-based Adaptive Structured Pruning for Large Language Models
> >
> > [2] Search for Efficient Large Language Models
> >
> > [3] LLM-Pruner: On the Structural Pruning of Large Language Models
> >
> > [4] SliceGPT: Compress Large Language Models by Deleting Rows and Columns
> >
> > [5] A Simple and Effective Pruning Approach for Large Language Models

---

> > > ### Author Response · Authors · 2024-11-26
> > > **Response to Reviewer 6JxQ (4)**
> > >
> > > **4. It is a better way for us to adopt the non-backward method for LLMs to recover model performance.**
> > >
> > > We agree with your suggestion that adopting non-backward methods for recovering model performance can be a promising direction, particularly as they generalize well to extremely large models while saving computational resources. In line with this perspective, we have conducted experiments to evaluate the performance of pruned models **without fine-tuning** and directly compared these results with existing methods such as SLEB, which also do not rely on fine-tuning for recovery (mentioned by Reviewer uw4p). Specifically, we conduct additional experiments on Llama-3.1-8B-It and Llama-3-8B  and compare our method with SLEB. As shown in the Table below,  the Reverse-order method consistently outperforms SLEB even without fine-tuning, further highlighting its effectiveness.
> > >
> > > |          Model          |          Method          |    PIQA    | HellaSwag  | OpenbookQA |   ARC-e   |   ARC-c   |   MMLU    |  CMMLU    | WinoGrande | Avg Acc |
> > > |:------------------------:|:------------------------:|:----------:|:----------:|:----------:|:---------:|:---------:|:---------:|:---------:|:----------:|:-------:|
> > > | Llama-3.1-8B-It | SLEB   | 0.7252±0.0104 | 0.4415±0.0050 | 0.2380±0.0191 | 0.6423±0.0098 | 0.3166±0.0136 | 0.3396±0.0040 | 0.2756±0.0042 | 0.5888±0.0138 | 0.4192  |
> > > | Llama-3.1-8B-It | Reverse-order   | 0.7002±0.0107 | 0.4021±0.0049 | 0.2920±0.0204 | 0.6178±0.0100 | 0.3993±0.0143 | 0.6346±0.0039 | 0.5458±0.0045 | 0.6251±0.0136 | 0.5271  |
> > > | Llama-3-8B | SLEB        | 0.7111±0.0106 | 0.4401±0.0050 | 0.2280±0.0188 | 0.6014±0.0100 | 0.2807±0.0131 | 0.2674±0.0037 | 0.2502±0.0040 | 0.5683±0.0139 | 0.3689  |
> > > | Llama-3-8B | Reverse-order        | 0.6921±0.0108 | 0.4035±0.0049 | 0.3040±0.0206 | 0.6014±0.0100 | 0.3720±0.0141 | 0.5603±0.0040 | 0.4216±0.0045 | 0.5975±0.0138 | 0.4940   |

---

> > > > ### Comment · Reviewer_6JxQ · 2024-11-26
> > > > **Follow up (2)**
> > > >
> > > > 4. I am not sure what is the sparsity of the provided results, if it is still the 6.3B model (pruned from 8B LLaMA-3 with 62.99% average accuracy), I think the results are not that good as the total sparsity is only around 22%. While for 20% structured pruning, as shown in Table 2 in work [1], the Wanda, FLAP and LLM-Pruner do not decrease much in average accuracy on zero-shot datasets.
> > > >
> > > >
> > > > [1] Fluctuation-based Adaptive Structured Pruning for Large Language Models
> > > >
> > > > [2] Search for Efficient Large Language Models
> > > >
> > > > [3] LLM-Pruner: On the Structural Pruning of Large Language Models
> > > >
> > > > [4] SliceGPT: Compress Large Language Models by Deleting Rows and Columns
> > > >
> > > > [5] A Simple and Effective Pruning Approach for Large Language Models

---

> > > > > ### Author Response · Authors · 2024-11-28
> > > > > **Response to Reviewer 6JxQ (5)**
> > > > >
> > > > > We sincerely appreciate the reviewer’s thoughtful feedback and the opportunity to further clarify the contributions and positioning of our work.
> > > > >
> > > > > **On the Suggested 6.3B Model.**
> > > > > The 6.3B model presented in our paper is selected as an illustrative example of the effectiveness of reverse-order pruning. Our choice is not intended to imply that this is the sole or universally optimal configuration. Instead, we aimed to demonstrate that the proposed method could achieve significant model compression while maintaining competitive performance.
> > > > >
> > > > > **Emphasizing the importance of avoiding fine-tuning.** As mentioned in our earlier response (**Response to Reviewer 6JxQ (4)**), we have conducted experiments specifically designed to evaluate the performance of pruned models without fine-tuning. In these experiments, we directly compared our Reverse-order pruning method with advanced approaches such as SLEB, which similarly avoid fine-tuning for recovery.  Experiments demonstrate that the Reverse-order pruning method consistently outperforms SLEB **even without fine-tuning** on Llama-3.1-8B-It and Llama-3-8B, further highlighting its effectiveness.
> > > > >
> > > > >
> > > > > We encourage the reviewer to refer to **our earlier response** and the accompanying results table for more detailed insights.
> > > > >
> > > > > We agree that pruning models must prioritize generalization. However, we respectfully disagree with the notion that “While other pruning methods [1] [2] [3] [4] [5] may rely on calibration samples to determine the importance of weights for pruning, they are often better at preserving the original model's performance,” How can we ensure that the model obtained by pruning on certain calibration samples can be generalized to other datasets? Besides, [6] has demonstrated that **the use of different calibration datasets can result in non-negligible variations in model performance**.
> > > > >
> > > > > **Comparison to those fine-grained pruning methods.**  As mentioned in our earlier response (2. Focus on layer pruning rather than channel pruning or weight pruning. in **Response to Reviewer 6JxQ (3)**), reverse-order pruning is **fundamentally different** in that it focuses on coarse-grained layer pruning rather than fine-grained pruning [1] [2] [3] [4] [5] of channels or weights. Layer pruning removes entire layers, resulting in a simpler and more computationally efficient model reduction. On the other hand, channel and weight pruning target specific components within layers, requiring more granular adjustments and potentially leading to greater computational overhead.
> > > > >
> > > > > We believe that **directly comparing reverse-order pruning to these finer-grained pruning methods is not entirely fair**, as they address different pruning objectives. While both approaches aim to reduce model size, layer pruning emphasizes simplicity and computational efficiency, which aligns with our goal of presenting a method that can be easily implemented in real-world scenarios with minimal computational resources.
> > > > >
> > > > > We hope this clarification addresses the reviewer’s concern and better articulates the difference between the objectives and complexities of our method versus the fine-grained pruning approaches.
> > > > >
> > > > > [1] Fluctuation-based Adaptive Structured Pruning for Large Language Models
> > > > >
> > > > > [2] Search for Efficient Large Language Models
> > > > >
> > > > > [3] LLM-Pruner: On the Structural Pruning of Large Language Models
> > > > >
> > > > > [4] SliceGPT: Compress Large Language Models by Deleting Rows and Columns
> > > > >
> > > > > [5] A Simple and Effective Pruning Approach for Large Language Models
> > > > >
> > > > > [6] Williams, Miles, and Nikolaos Aletras. "On the impact of calibration data in post-training quantization and pruning." Proceedings of the 62nd Annual Meeting of the Association for Computational Linguistics (Volume 1: Long Papers). 2024.

---

> > > > > > ### Comment · Reviewer_6JxQ · 2024-11-28
> > > > > > **Follow up for response (5)**
> > > > > >
> > > > > > 1. I believe the 6.3B model shows some limitations, primarily due to the lack of flexibility in your method. It is beneficial to explore and provide results with a broader range of sparsity ratios to enhance adaptability and performance insights.
> > > > > >
> > > > > > 2.a. My concern about the accuracy obtained without fine-tuning is not about comparing it to other methods but rather about its performance relative to fine-grained pruning methods that do not adopt fine-tuning. Specifically, I noticed that your method results in a significant accuracy drop compared to the dense model, which raises questions about its effectiveness in this context (Llama-3.1-8B-It: 62.99% -> 52.71%; LLaMA3-8B: 60.93% -> 49.4%).
> > > > > >
> > > > > > 2.b. The main point I want to emphasize is that further fine-tuning can diminish the original potential of LLMs, as the quality of the fine-tuning dataset often falls short when compared to the pretraining dataset.
> > > > > >
> > > > > > 2.c. Regarding the generalization of calibration, it is acknowledged that there is some bias toward the dataset. However, based on the ablation study results presented in Table 7 of work [1] and Table A2 of work [2], the generalization remains relatively robust, as evidenced by the minimal variation in accuracy across different datasets. This suggests that the calibration approach retains consistent performance despite the inherent dataset-specific biases.
> > > > > >
> > > > > > 3. I acknowledge that the layer pruning is different from the fine-grained pruning methods. However, my main point is, according to the analysis in [1] [3], there is redundant layers in the front layers of LLMs. I think you should give more results about this. For example, for around 20% sparsity in 7B (8B) model which contains only 32 layers, it will not take much resource to search for the optimal strategy of pruning 6 layers (or 5 layers) in the 32 layers. This kind of ablation study is necessary to show the effectiveness of your conclusion on the reserve-pruning.
> > > > > >
> > > > > > 4. As for the computational resources, those fine-grained methods [1] [2] [3] [4] [5], which do not adopt fine-tuning or even weight update, do not require the backward progress, which saves more resources than the methods that require fine-tuning. Even the work [2], which adopts the search for LLMs, can achieve comparable efforts to those pruning works [1] [3] [4]. Thus, I do not think the computational resources is kind of overhead to those methods.
> > > > > >
> > > > > > [1] Fluctuation-based Adaptive Structured Pruning for Large Language Models
> > > > > >
> > > > > > [2] Search for Efficient Large Language Models
> > > > > >
> > > > > > [3] LLM-Pruner: On the Structural Pruning of Large Language Models
> > > > > >
> > > > > > [4] SliceGPT: Compress Large Language Models by Deleting Rows and Columns
> > > > > >
> > > > > > [5] A Simple and Effective Pruning Approach for Large Language Models

---

> > > ### Comment · Reviewer_6JxQ · 2024-11-26
> > > **Follow up (1)**
> > >
> > > Hi, thanks for your reply and explanation.
> > >
> > > 1. The reason I reference works [1] and [2] is to suggest that you take the global optimum into account. Given the relatively small search space involved, especially in the early layers, this consideration is crucial. For instance, in models with 7B or 8B parameters, there are only 32 layers. If the importance of each of these 32 layers can be thoroughly examined, it would allow for the creation of smaller models with a broader range of sparse ratios, enabling greater flexibility and efficiency. However, in your paper, there is only 6.3B models you suggested as the optimal one.
> > >
> > > 2.a. I disagree with this point. While other pruning methods [1] [2] [3] [4] [5] may rely on calibration samples to determine the importance of weights for pruning, they are often better at preserving the original model's performance, which stems from the extensive and powerful pretraining dataset. Further fine-tuning, on the other hand, can potentially compromise the model's generalization ability. Therefore, I believe models that avoid additional fine-tuning are more likely to achieve superior generalization compared to those that incorporate it.
> > >
> > > 2.b. As for the comparison to those fine-grained pruning methods [1] [2] [3] [4] [5], they can also generate the small dense models which are practical in real-world scenarios. And their methods are general and can be applied to other LLMs easily as the cost of those methods are not that much especially when compared to the fine-tuning on Alpaca.
> > >
> > > 2.c. According to the Table 2 in FLAP, with 20% structured sparsity, FLAP achieves a little drop on zero-shot performance, and Wanda (which does not require weight update after pruning) even achieves better performance than dense model. On the other hand, according to the non-finetuning results and the ablation study for number of fine-tuning samples, your layer pruning method does not achieve good results and is sensitive to the number of fine-tuning samples.
> > >
> > > 3.a. I am afraid you misunderstand the Figure 3 of work [1], the figure shows the **magnitude of the metric** proposed by this work. rather than the magnitude of the weights with $l1$ or $l2$ norm. You can check how the metric is defined in Equation 5 of work [1].
> > >
> > > 3.b. Also, according to Figure 3 in the work [3] which shows the layer sensitivity for pruning, it shows that the middle layers are redundant while the latest layers are sensitive.
> > >
> > >
> > >
> > > [1] Fluctuation-based Adaptive Structured Pruning for Large Language Models
> > >
> > > [2] Search for Efficient Large Language Models
> > >
> > > [3] LLM-Pruner: On the Structural Pruning of Large Language Models
> > >
> > > [4] SliceGPT: Compress Large Language Models by Deleting Rows and Columns
> > >
> > > [5] A Simple and Effective Pruning Approach for Large Language Models

---

### Official Review · Reviewer_MhNQ · 2024-11-03

**Soundness:** 3
**Presentation:** 4
**Contribution:** 3
**Rating:** 6
**Confidence:** 5

**Summary:**

In this work, the authors conduct comprehensive empirical study on layer-wise post-training pruning across various LLMs. Specifically, they present three key conclusion: (1) reverse-order layer pruning outperforms other layer-wise pruning importance metrics, (2) fine-tuning the last few remaining layers yields better performance than LoRA, and (3) iterative layer pruning shows no advantage over one-shot layer pruning. Based on these analysis, the authors develop pruned models using Llama-3.1-Instruct, achieving better performance compared to other LLMs of the same or larger size.

**Strengths:**

1. The paper is well organized and easy to follow.
2. It‘s inspired to see fine-tuning last few remaining layers (e.g.,  3) can outperform LoRA.
3. The pruned model based on LLama-3.1-Instruct shows better performance compared prior LLMs with similar model size.

**Weaknesses:**

1. Similar conclusions to Insight #1 and Insight #3 have been noted in prior work. Specifically, [1] demonstrates that deeper layers are less effective, so it would be helpful to clarify how this work differs from [1]. Additionally, as the authors mentioned, [2] shows that iterative pruning provides no added benefit.

2. The authors only present the results of the pruned model after fine-tuning. It would be informative to see the results prior to fine-tuning to see if the proposed method consistently outperforms others.

3. It would also be valuable to test the proposed method on the OPT model. As revealed in [3], unlike other LLMs, OPT models exhibit high redundancy in shallow layers rather than in deeper layers by using cosine similarity analysis.

[1] Gromov, Andrey, et al. "The unreasonable ineffectiveness of the deeper layers." arXiv preprint arXiv:2403.17887 (2024).

[2]  Compact language models via pruning and knowledge distillation. arXiv preprint arXiv:2407.14679, 2024

[3] Chen, Xiaodong, Yuxuan Hu, and Jing Zhang. "Compressing large language models by streamlining the unimportant layer." arXiv preprint arXiv:2403.19135

**Questions:**

Overall, I find this work valuable for offering new insights into post-training layer-wise pruning, particularly with Insight #2. However, the work could be strengthened by addressing the questions as shown in Weakness above: (1) clarify the differences compared to [1], (2) analyze performance both before and after fine-tuning, and (3) evaluate the proposed method on the OPT model.

**Details Of Ethics Concerns:**

No ethics concerns found.

---

> ### Author Response · Authors · 2024-11-25
> **Response to Reviewer MhNQ**
>
> **Similar conclusions to Insight #1 and Insight #3.**
>
> For Insight #1, authors in [1] find that removing layers beginning at the penultimate layer and proceeding from deep to shallow until the desired number of layers have been removed is effective. In contrast, reverse-order pruning directly cuts off the last few layers without retaining the last layer, which is different from [1]. Besides, we propose to use partial-layer fine-tuning in LLMs rather than the popular LoRA methods, i.e., freezing the other layers and fine-tuning only the last few remaining layers and lm_head. To the best of our knowledge, we are the first one demonstrating the effectiveness of partial-layer fine-tuning in LLM pruning.
>
> For Insight #3：Thank you for your feedback. We acknowledge the insights from [2], and indeed, similar conclusions about iterative pruning provide no added benefit. However, we would like to clarify that the context of our work differs from that of [2]. While [2] demonstrates that iterative pruning is ineffective when using knowledge distillation to retrain the pruned model, our experiments focus on performance recovery of iterative pruning using partial layer fine-tuning and LoRA. As mentioned in Table 5 of our initial submission, Iterative pruning offers no benefit when using LoRA or partial-layer fine-tuning to restore the pruned model performance. It is worth noting that fine-tuning using knowledge distillation is a full-model fine-tuning method, while the LoRA and partial-layer fine-tuning we use are partial-model fine-tuning methods. Therefore, our conclusions are orthogonal to those of [2] and complement each other.
>
> **Present the results of the pruned model prior to fine-tuning.**
> Thank you for this insightful suggestion. We conduct additional experiments and compare our method with SLEB [3], which is an advanced training-free layer pruning method (mentioned by Reviewer uw4p) on the Table below. The result demonstrates that our proposed method consistently outperforms SLEB even without fine-tuning, further highlighting its effectiveness.
> We appreciate your suggestion, as it has helped enhance the comprehensiveness of our evaluation.
>
> |          Model          |          Method          |    PIQA    | HellaSwag  | OpenbookQA |   ARC-e   |   ARC-c   |   MMLU    |  CMMLU    | WinoGrande | Avg Acc |
> |:------------------------:|:------------------------:|:----------:|:----------:|:----------:|:---------:|:---------:|:---------:|:---------:|:----------:|:-------:|
> | Llama-3.1-8B-It | SLEB   | 0.7252±0.0104 | 0.4415±0.0050 | 0.2380±0.0191 | 0.6423±0.0098 | 0.3166±0.0136 | 0.3396±0.0040 | 0.2756±0.0042 | 0.5888±0.0138 | 0.4192  |
> | Llama-3.1-8B-It | Reverse-order   | 0.7002±0.0107 | 0.4021±0.0049 | 0.2920±0.0204 | 0.6178±0.0100 | 0.3993±0.0143 | 0.6346±0.0039 | 0.5458±0.0045 | 0.6251±0.0136 | 0.5271  |
> | Llama-3-8B | SLEB        | 0.7111±0.0106 | 0.4401±0.0050 | 0.2280±0.0188 | 0.6014±0.0100 | 0.2807±0.0131 | 0.2674±0.0037 | 0.2502±0.0040 | 0.5683±0.0139 | 0.3689  |
> | Llama-3-8B | Reverse-order        | 0.6921±0.0108 | 0.4035±0.0049 | 0.3040±0.0206 | 0.6014±0.0100 | 0.3720±0.0141 | 0.5603±0.0040 | 0.4216±0.0045 | 0.5975±0.0138 | 0.4940   |

---

> ### Author Response · Authors · 2024-11-25
> **Response to Reviewer MhNQ (2)**
>
> **Test the proposed method on the OPT model.** Thank you for this valuable suggestion. Following your suggestion, we conduct experiments on OPT-6.7B and Llama2-7B using reverse-order pruning and compare it with [4]. As suggested by the cosine similarity analysis in Figure 2 of [4], OPT-6.7B exhibits high redundancy in shallow layers and Llama2-7B exhibits high redundancy in deep layers. Therefore, we reproduce the method proposed in [4] (Method: Cos) by pruning 2-9th layers for OPT-6.7B, the 22-29th layers for Llama2-7B and fine-tune the pruned models with LoRA on Alpaca-cleaned dataset. For our method, we just prune the last 8 layers in OPT-6.7B and Llama2-7B, freeze the other layers and fine-tune only the last 3  layers and lm_head on Alpaca-cleaned dataset.
>
> From the comparisons on OPT and Llama2-7B, we find pruning the shallow layers of the OPT model is overall better than reverse-order pruning, while the performance improvement is not always consistent as shown on MMLU, CMMLU, and WinoGrande where Cos works slightly worse than reverse-order. We believe the metric in [4] did reveal the redundancy of shallow layers in OPT but the detailed layer index may not always be accurate. For example, [4] suggests pruning the 22-29th layers while experiments in the table below shows pruning the last 8 layers is overall better.
>
> To further demonstrate our effectiveness, we conduct experiments on Llama-3.1-8B-It. As shown in the table, our pruned model achieves an average accuracy of 0.5807, far exceeding the cos method 0.2633. We has cited the paper and added the discussion in our revised version.
>
> |  Model    |  Method  |    PIQA    |  HellaSwag  |  OpenbookQA  |  ARC-e    |  ARC-c    |  MMLU     |  CMMLU    |  WinoGrande  | Avg Acc |
> |:-----------:|:----------:|:-----------:|:-----------:|:-----------:|:-----------:|:-----------:|:-----------:|:-----------:|:-----------:|:-----------:|
> | Llama2-7B | Reverse-order  | 0.7089±0.0106 | 0.4875±0.0050 | 0.3020±0.0206  | 0.6317±0.0099 | 0.3780±0.0142 | 0.2789±0.0038 | 0.2590±0.0041 | 0.6251±0.0136  | 0.4589      |
> |      Llama2-7B         |  Cos     | 0.7301±0.0104 | 0.4904±0.0050 | 0.2640±0.0197  | 0.6334±0.0099 | 0.3311±0.0138 | 0.2599±0.0037 | 0.2478±0.0040 | 0.6748±0.0132  | 0.4242      |
> |  OPT-6.7B  |  Reverse-order   | 0.6893±0.0108 | 0.4068±0.0049 | 0.2180±0.0185  | 0.4949±0.0103 | 0.2730±0.0130 | 0.2469±0.0036 | 0.2526±0.0040 | 0.6101±0.0137  | 0.3717      |
> |     OPT-6.7B           |  Cos     | 0.7209±0.0105 | 0.4439±0.0050 | 0.2500±0.0194  | 0.5795±0.0101 | 0.2969±0.0134 | 0.2388±0.0036 | 0.2500±0.0040 | 0.5888±0.0138  | 0.4211      |
> |  Llama-3.1-It   |  Reverse-order  | 0.7383±0.0103 | 0.5323±0.0050 | 0.3080±0.0207 | 0.7260±0.0092 | 0.4684±0.0146 | 0.6567±0.0038 | 0.5515±0.0045 | 0.6646±0.0133 | 0.5807 |
> |  Llama-3.1-It  |  Cos  | 0.5773±0.0115 | 0.2878±0.0045 | 0.1520±0.0161 | 0.3674±0.0099 | 0.1706±0.0110 | 0.2342±0.0036 | 0.2466±0.0040 | 0.5036±0.0141 | 0.3174 |
>
> [1] Gromov, Andrey, et al. "The unreasonable ineffectiveness of the deeper layers." arXiv preprint arXiv:2403.17887 (2024).
>
> [2] Muralidharan, Saurav, et al. "Compact language models via pruning and knowledge distillation." The Thirty-eighth Annual Conference on Neural Information Processing Systems. 2024.
>
> [3] Song, Jiwon, et al. "SLEB: Streamlining LLMs through Redundancy Verification and Elimination of Transformer Blocks." arXiv preprint arXiv:2402.09025 (2024).
>
> [4] Chen, Xiaodong, Yuxuan Hu, and Jing Zhang. "Compressing large language models by streamlining the unimportant layer." arXiv preprint arXiv:2403.19135

---

> > ### Author Response · Authors · 2024-11-28
> > **A Kind Reminder for Reviewer MhNQ**
> >
> > Dear Reviewer,
> >
> > Thank you for your valuable feedback and thorough review of our paper. Your insights have greatly contributed to refining our work. In response to the specific concerns you raised, we have provided detailed explanations to address each concern comprehensively. Below, we summarize your concerns and our key responses:
> >
> > * **[W1: similar conclusions to Insight #1 and Insight #3.]:** We differentiate our method, reverse-order pruning, from Insight #1 by **directly removing the last few layers**, rather than starting from the penultimate layer and proceeding from deep to shallow. We introduce partial-layer fine-tuning in LLM pruning, where only the remaining layers and lm_head are fine-tuned, unlike the more commonly used LoRA methods.
> > We emphasize that fine-tuning via knowledge distillation is a full-model process, while LoRA and partial-layer fine-tuning are partial-model methods. Our findings are **complementary to Insight #3**.
> > * **[W2: the results of the pruned model prior to fine-tuning.]:** We conduct additional experiments to present the pruned model’s performance without fine-tuning, comparing our method to SLEB, a training-free layer pruning method. Our method **consistently outperforms SLEB** even without fine-tuning, which further highlights its effectiveness.
> >
> > * **[W3: Test the proposed method on the OPT model.]:** We test the reverse-order pruning method on OPT-6.7B, Llama2-7B and Llama-3.1-8B-It, which outperforms the Cos method.
> >
> > We have incorporated your valuable suggestions into the revised manuscript. Thank you once again for your insightful feedback!
> >
> > If our rebuttal has adequately addressed your concerns, we kindly request that you consider revising your score accordingly. An increased score is critically important to our work at this stage.
> >
> > We remain open and glad to any additional questions or feedback you may have. Your efforts and detailed review are greatly appreciated, and we value the opportunity to improve our work based on your input. Thank you once again for your time and consideration. We look forward to your further feedback.
> >
> > Best regards,
> >
> > Authors of Paper 137

---

> ### Comment · Area_Chair_caTz · 2024-11-30
>
> Dear Reviewer,
>
> Could you kindly respond and indicate whether authors have addressed your concerns?
>
> Thanks, AC

---

### Official Review · Reviewer_uw4p · 2024-11-03

**Soundness:** 3
**Presentation:** 3
**Contribution:** 2
**Rating:** 3
**Confidence:** 5

**Summary:**

This paper explores layer pruning in Large Language Models (LLMs) to reduce computational overhead while maintaining performance. The authors conduct extensive experiments across multiple dimensions, including different layer selection metrics, fine-tuning methods, and pruning strategies. Their findings suggest that a simple reverse-order pruning strategy—pruning the last 25% of layers—performs as well as more sophisticated methods. Applying these insights, they prune Llama-3.1-8B-Instruct to create Llama-3.1-6.3B-It models, which outperform several popular LLMs of similar size.

**Strengths:**

1. The paper conducts lots of empirical study to support their findings, which may be beneficial for the community for future research.

2. By releasing the pruned model weights and code, the authors contribute to open science and facilitate reproducibility and further research in the field.

**Weaknesses:**

1. The main findings emphasize that simple methods can be highly effective. While valuable, this insight may be seen as incremental, confirming existing intuitions rather than introducing new methodologies. The effectiveness of pruning the last layers and fine-tuning only specific parts of the model aligns with established practices in model compression and transfer learning.

2. The study compares simple pruning metrics with Block Influence (BI) from ShortGPT [1] but does not include comparisons with more recent and advanced layer pruning methods such as SLEB [2] and FinerCut [3]. Both SLEB and FinerCut have introduced innovative approaches to layer pruning in LLMs, offering potentially significant improvements in efficiency and performance.

3. Although the paper finds that reverse-order pruning (pruning the last several layers) is effective, it does not delve into why this method outperforms other metrics. An analysis of the role and importance of the last layers in LLMs could provide valuable insights and contribute to the development of more effective pruning strategies.

[1] Men, Xin, et al. "Shortgpt: Layers in large language models are more redundant than you expect." arXiv preprint arXiv:2403.03853 (2024).

[2] Song, Jiwon, et al. "SLEB: Streamlining LLMs through Redundancy Verification and Elimination of Transformer Blocks." arXiv preprint arXiv:2402.09025 (2024).

[3] Zhang, Yang, et al. "FinerCut: Finer-grained Interpretable Layer Pruning for Large Language Models." arXiv preprint arXiv:2405.18218 (2024).

**Questions:**

1.  Despite the authors' efforts to provide code and models, some experimental details are insufficiently specified. For example, exact hyperparameters for all experiments, detailed configurations of the fine-tuning setup, and the procedures for selecting and processing calibration samples for the data-driven pruning metrics are not fully described.  Can the authors give more details?

---

> ### Author Response · Authors · 2024-11-25
> **Response to Reviewer uw4p**
>
> **The insight may be seen as incremental.** While we agree that fine-tuning specific layers is a well-established practice in model compression and transfer learning on CNNs, we would like to emphasize that our work is, to the best of our knowledge, the first to propose and demonstrate the effectiveness of this approach in the context of pruning large language models (LLMs). As mentioned in Table 2 of our initial submission, our results highlight that fine-tuning only the last few remaining layers and lm_head achieves significant improvements in both performance and efficiency, particularly when compared to the popular LoRA fine-tuning. Notably, our findings highlight a unique advantage of LLM layer pruning, as LoRA and partial-layer fine-tuning exhibit comparable performance during full-model fine-tuning (as shown in Table 3 of our initial submission). This distinction offers new insights and practical contributions to the field of LLM pruning.
>
>  We appreciate your recognition of the value of our findings and hope this clarification underscores the novelty of our work.
>
> **Comparisons with more recent and advanced layer pruning methods.** Thank you for pointing out the importance of comparing with more advanced layer pruning methods such as SLEB and FinerCut. While we recognize the contributions of FinerCut, its implementation is not publicly available, which prevents us from conducting a fair and reproducible comparison. To this end, we conduct experiments with SLEB on Llama-3.1-8B-It and Llama3-8B. For each pruning method, we prune 8 layers. Since SLEB is processed based on the inference-only approach without any finetuning, for a fair comparison, we only prune the model with the last 8 layers without finetuning. As shown in Table below, reverse-order pruning outperforms SLEB, which demonstrates its efficiency.
> |          Model          |          Method          |    PIQA    | HellaSwag  | OpenbookQA |   ARC-e   |   ARC-c   |   MMLU    |  CMMLU    | WinoGrande | Avg Acc |
> |:------------------------:|:------------------------:|:----------:|:----------:|:----------:|:---------:|:---------:|:---------:|:---------:|:----------:|:-------:|
> | Llama-3.1-8B-It | SLEB   | 0.7252±0.0104 | 0.4415±0.0050 | 0.2380±0.0191 | 0.6423±0.0098 | 0.3166±0.0136 | 0.3396±0.0040 | 0.2756±0.0042 | 0.5888±0.0138 | 0.4192  |
> | Llama-3.1-8B-It | Reverse-order   | 0.7002±0.0107 | 0.4021±0.0049 | 0.2920±0.0204 | 0.6178±0.0100 | 0.3993±0.0143 | 0.6346±0.0039 | 0.5458±0.0045 | 0.6251±0.0136 | 0.5271  |
> | Llama-3-8B | SLEB        | 0.7111±0.0106 | 0.4401±0.0050 | 0.2280±0.0188 | 0.6014±0.0100 | 0.2807±0.0131 | 0.2674±0.0037 | 0.2502±0.0040 | 0.5683±0.0139 | 0.3689  |
> | Llama-3-8B | Reverse-order        | 0.6921±0.0108 | 0.4035±0.0049 | 0.3040±0.0206 | 0.6014±0.0100 | 0.3720±0.0141 | 0.5603±0.0040 | 0.4216±0.0045 | 0.5975±0.0138 | 0.4940   |
>
> Besides, to evaluate the effectiveness of partial-layer fine-tuning, we freeze the other layers and fine-tune only the last three remaining layers and lm_head using both Alpaca-cleaned and Dolly datasets. For SLEB, we use the same parameter settings as for partial-layer fine-tuning to perform LoRA fine-tuning on the Alpaca-cleaned dataset. The experimental results demonstrate that fine-tuning specific layers after pruning with the reverse-order  method yields significantly better performance compared to fine-tuning with LoRA using the existing SLEB method.
>
> |          Model         |          Method          |          Dataset         |    PIQA    | HellaSwag | OpenbookQA |   ARC-e   |   ARC-c   |   MMLU    |   CMMLU   | WinoGrande | Avg Acc |
> |:------------------------:|:------------------------:|:------------------------:|:----------:|:---------:|:----------:|:---------:|:---------:|:---------:|:---------:|:----------:|:-------:|
> | Llama-3.1-8B-It | SLEB  | Alpaca-cleaned| 0.7573±0.0100 | 0.4973±0.0050 | 0.2680±0.0198 | 0.6970±0.0094 | 0.3865±0.0142 | 0.4305±0.0041 | 0.3338±0.0044 | 0.6385±0.0135 | 0.5011  |
> | Llama-3.1-8B-It | Reverse-order  | Alpaca-cleaned  | 0.7383±0.0103 | 0.5323±0.0050 | 0.3080±0.0207 | 0.7260±0.0092 | 0.4684±0.0146 | 0.6567±0.0038 | 0.5515±0.0045 | 0.6646±0.0133 | 0.5807  |
> | Llama-3.1-8B-It | Reverse-order  | Dolly  | 0.7709±0.0098 | 0.5541±0.0050 | 0.3000±0.0205 | 0.7424±0.0090 | 0.4838±0.0146 | 0.6753±0.0038 | 0.5522±0.0045 | 0.7032±0.0128 | 0.5977  |
> | Llama-3-8B  | SLEB  | Alpaca-cleaned          | 0.7514±0.0101 | 0.5026±0.0050 | 0.2780±0.0201 | 0.7071±0.0093 | 0.3720±0.0141 | 0.3115±0.0039 | 0.2683±0.0041 | 0.5967±0.0138 | 0.3947  |
> | Llama-3-8B |  Reverse-order | Alpaca-cleaned        | 0.7388±0.0102 | 0.5476±0.0050 | 0.3160±0.0208 | 0.7218±0.0092 | 0.4394±0.0145 | 0.6179±0.0038 | 0.4497±0.0045 | 0.6748±0.0132 | 0.5633  |
> | Llama-3-8B| Reverse-order | Dolly         | 0.7274±0.0104 | 0.5123±0.0050 | 0.3040±0.0206 | 0.6721±0.0096 | 0.4172±0.0144 | 0.6186±0.0038 | 0.4622±0.0045 | 0.6811±0.0131 | 0.5494  |

---

> ### Author Response · Authors · 2024-11-25
> **Response to Reviewer uw4p (2)**
>
> **An analysis of the role and importance of the last layers in LLMs.** Thank you for your thoughtful feedback. After pruning, the reduced parameter set may necessitate task-specific representation learning to optimize performance. Fine-tuning the top layers (those closer to the output) and the lm_head proves particularly effective since these layers are most directly associated with task-specific features.
> Besides, fine-tuning the last layers is a widely recognized approach in transfer learning [1-3], further supporting our claim that last layers are directly associated with task-specific features.  LoRA, on the other hand, modifies parameters indirectly and may not achieve the same level of task-specific optimization.
>
> **Experimental details.** Thank you for your thoughtful feedback. We appreciate your attention to the experimental details and your interest in the reproducibility of our results. We set the batch size and epoch to 64 and 2, respectively. We use the AdamW optimizer. The learning rate is set to 1×10−5 with 100 warming steps. We encourage you to explore and experiment with the default settings in our codes. If there are still questions, we welcome additional questions or requests for clarification. We are committed to supporting reproducibility.
>
> [1] Zhuang, Fuzhen, et al. "A comprehensive survey on transfer learning." Proceedings of the IEEE 109.1 (2020): 43-76.
>
> [2] Jang Y, Lee H, Hwang S J, et al. Learning what and where to transfer[C]//International conference on machine learning. PMLR, 2019: 3030-3039.
>
> [3] Huh, Minyoung, Pulkit Agrawal, and Alexei A. Efros. "What makes ImageNet good for transfer learning?." arXiv preprint arXiv:1608.08614 (2016).

---

> > ### Comment · Reviewer_uw4p · 2024-11-26
> >
> > Thank you for the authors' response and the additional experiments provided. While the results demonstrate effectiveness, I remain concerned about the novelty and contribution of this work. Specifically, the paper appears to lack a clear and substantial contribution that could guide or inspire future research in the field.
> >
> > The paper is like "Shooting in the dark and pretending you aimed", where choosing the last layer and fine-tuning seems more like an empirical finding after extensive experimentation rather than a method driven by a well-defined motivation or theoretical insight. This raises concerns about the underlying rationale behind the proposed method.
> >
> > In research, we typically expect a strong contribution from one of two scenarios: (1) identifying a compelling problem or motivation and presenting effective solutions, or (2) uncovering an unusual or unexpected observation and conducting an in-depth analysis to understand its underlying causes. For this paper, the latter seems more applicable, yet the analysis provided does not feel sufficiently mature or comprehensive to support this claim.
> >
> > Additionally, I share Reviewer 6JxQ's concerns regarding the lack of clarity surrounding how the proposed method works. Furthermore, the reliance on specific datasets for fine-tuning appears to play a significant role in the observed accuracy improvements, which raises questions about the generalizability of the approach.

---

> > > ### Author Response · Authors · 2024-11-26
> > > **Response to Reviewer uw4p (3)**
> > >
> > > We appreciate the reviewer’s detailed feedback and their acknowledgment of the effectiveness of our findings. Below, we address the primary concerns raised and provide further clarifications.
> > >
> > > **1. Clear Aim and Motivation**
> > >
> > > We respectfully disagree with the characterization that our work is "shooting in the dark and pretending you aimed." As stated in our abstract, our work seeks to answer an important and practical question: What are the best practices for layer pruning in large language models (LLMs)? While there exist many sophisticated layer pruning strategies, often requiring complex metrics or extra validation data, their true effectiveness remains unclear. Our goal is to explore whether a simple and effective alternative exists. Thus, the aim of this work is not only clear but also highly relevant to the community.
> > >
> > > **2. Regarding the Two Scenarios of Contribution**
> > >
> > > We address the two scenarios identified by the reviewer:
> > > * **Compelling Problem or Motivation and Effective Solutions.** The problem is clearly stated in our abstract: What are the best practices for layer pruning in LLMs? This is a compelling problem because, despite the growing importance of model compression, the best practices for layer pruning remain unknown. Our results, acknowledged as effective in the review (e.g., "reverse-order pruning... is effective"), confirm that our work addresses this problem successfully.
> > >
> > > * **Unusual or Unexpected Observation and In-Depth Analysis.** The effectiveness of reverse-order pruning and partial-layer fine-tuning is indeed "unusual" within the context of existing literature. The common expectation is that more sophisticated strategies would outperform simpler approaches, yet our findings challenge this assumption. Specifically:
> > >
> > >     * **Reverse-Order Pruning:** Our analysis reveals that pruning the final layers, which are more task-specific and prone to overfitting, retains the general embedding extraction capabilities crucial for broader generalization. This insight aligns with the understanding of how LLMs manage task-specific versus general representations.
> > >
> > >     * **Partial-Layer Fine-Tuning:** We show that fine-tuning the final layers and the lm_head is particularly effective because these layers are closest to the output and most associated with task-specific features. This approach aligns with widely recognized practices in transfer learning [1-3]. In contrast, methods like LoRA modify parameters indirectly, which may limit their ability to achieve the same level of task-specific optimization.
> > > These explanations, as reiterated in our previous rebuttal, provide a theoretical grounding for our empirical observations.
> > >
> > > **3. Generalization and Dataset Dependence**
> > >
> > > Regarding concerns about the generalization of our findings, we note that our fine-tuning experiments used Alpaca and Dolly datasets, both of which are widely adopted and not specifically designed for our test datasets (e.g., PIQA, HellaSwag, OpenbookQA, ARC-e, ARC-c, MMLU, CMMLU, and WinoGrande). The numerical results across these diverse datasets consistently demonstrate strong generalization, addressing concerns about reliance on specific datasets. Besides, we have conducted experiments to evaluate the performance of pruned models **without fine-tuning on any datasets** and directly compared these results with SLEB (see the “**Comparisons with more recent and advanced layer pruning methods.**” of **Response to Reviewer uw4p**).
> > >
> > > We hope this clarifies our contributions, addresses concerns about the rationale and theoretical insight behind our findings, and demonstrates the novelty and generalizability of our findings.
> > >
> > > [1] Zhuang, Fuzhen, et al. "A comprehensive survey on transfer learning." Proceedings of the IEEE 109.1 (2020): 43-76.
> > >
> > > [2] Jang Y, Lee H, Hwang S J, et al. Learning what and where to transfer[C]//International conference on machine learning. PMLR, 2019: 3030-3039.
> > >
> > > [3] Huh, Minyoung, Pulkit Agrawal, and Alexei A. Efros. "What makes ImageNet good for transfer learning?." arXiv preprint arXiv:1608.08614 (2016).

---

### Official Review · Reviewer_Dtqt · 2024-11-04

**Soundness:** 3
**Presentation:** 3
**Contribution:** 3
**Rating:** 6
**Confidence:** 3

**Summary:**

In this paper, the author spent thousands of GPU hours to reassess the practices and insights of layer pruning in LLMs. The results showed that reverse-order pruning is simple yet effective (simply pruning the last several layers performs better than many complex pruning metrics); partial-layer fine-tuning (freezing the other layers and fine-tuning only the last few remaining layers and lm_head) can achieve higher accuracy than LoRA fine-tuning; one-shot pruning in more beneficial than iterative fine-tuning considering both training costs and performance gains.

**Strengths:**

+ The structure of the paper is well-designed and organized
+ The background information is rich, especially those math equations, which makes it easy for someone who is not familiar with this field to understand the concept of layer pruning and relevant techniques.
+ Because the paper is an experiments-based publication, it is very good there are lots of diverse experiments conducted in the paper, including plenty of different datasets and models.
+ The results of the experiment are very rich in graphs and tables, and the results are clear briefly.

**Weaknesses:**

- Some of the metrics don’t have a mark to indicate whether the lower or the higher it is, the performance is better, especially some uncommon metrics.

**Questions:**

Please refer to the weakness part.

---

> ### Author Response · Authors · 2024-11-25
> **Response to Reviewer Dtqt**
>
> **Some of the metrics don’t have a mark to indicate whether the lower or the higher it is：**   We apologize for the lack of clarity in the original table, which might have confused the interpretation of certain metrics. To address this, we have revised the table to include explicit markers (e.g., ↑ or ↓) to indicate whether a higher or lower value is better for each metric. We sincerely thank the reviewer for pointing this out, as it has helped improve the clarity and presentation of our paper.

---

> > ### Author Response · Authors · 2024-11-28
> > **A Kind Reminder for Reviewer Dtqt**
> >
> > Dear Reviewer,
> >
> > Thank you for your valuable feedback and thorough review of our paper. Your insights have greatly contributed to refining our work. In response to the specific concerns you raised, we have provided detailed explanations to address each concern comprehensively. Below, we summarize your concerns and our key responses:
> >
> > * **[W1: Some of the metrics don’t have a mark to indicate whether the lower or the higher it is.]:** We greatly appreciate your constructive feedback, as well as your support and recommendation for acceptance. We will carefully incorporate the suggested descriptions of metrics into our revision to further strengthen the manuscript.
> >
> > We have incorporated your valuable suggestions into the revised manuscript. Thank you once again for your insightful feedback!
> >
> > If our rebuttal has adequately addressed your concerns, we kindly request that you consider revising your score accordingly. An increased score is critically important to our work at this stage.
> >
> > We remain open and glad to any additional questions or feedback you may have. Your efforts and detailed review are greatly appreciated, and we value the opportunity to improve our work based on your input. Thank you once again for your time and consideration. We look forward to your further feedback.
> >
> > Best regards,
> >
> > Authors of Paper 137

---

> ### Comment · Area_Chair_caTz · 2024-11-30
>
> Dear Reviewer,
>
> Could you kindly respond and indicate whether authors have addressed your concerns?
>
> Thanks, AC

---

### Comment · Area_Chair_caTz · 2024-11-28
**Reviewers, please kindly respond**

Dear Reviewers,

If you have not responded to author's rebuttal, please kindly do so as soon as possible. The deadline is Dec 2, but the authors can potentially further clarify questions if you respond earlier. Thanks!

Best, AC

---

### Note · Authors · 2024-12-10

I have read and agree with the venue's withdrawal policy on behalf of myself and my co-authors.